# MicroRNAs are minor constituents of extracellular vesicles that are rarely delivered to target cells

Manuel Albanese[1,2,3‡]*, Yen-Fu Adam Chen[1,3‡], Corinna Hüls[1,3], Kathrin Gärtner[1,3], Takanobu Tagawa[1,3¤], Ernesto Mejias-Perez[2,3], Oliver T. Keppler[2,3], Christine Göbel[1,3], Reinhard Zeidler[1,3,4], Mikhail Shein[5], Anne K. Schütz[5,6], Wolfgang Hammerschmidt[1,3]*

1 Research Unit Gene Vectors, Helmholtz Zentrum München, German Research Center for Environmental Health, Munich, Germany, 2 Max von Pettenkofer Institute and Gene Center, Virology, National Reference Center for Retroviruses, Faculty of Medicine, LMU München, Munich, Germany, 3 German Centre for Infection Research (DZIF), Partner site Munich, Germany, 4 Department of Otorhinolaryngology, Klinikum der Universität München, Munich, Germany, 5 Bavarian NMR Center, Department of Chemistry, Technical University of Munich, Garching, Germany, 6 Institute of Structural Biology, Helmholtz Zentrum München, Neuherberg, Germany

¤ Current address: HIV and AIDS Malignancy Branch, Center for Cancer Research, National Cancer Institute, National Institutes of Health, Bethesda, Maryland, United States of America
‡ These authors share first authorship on this work.
* albanese@ingm.org (MA); hammerschmidt@helmholtz-muenchen.de (WH)

**Data Availability Statement:** All relevant data are within the manuscript and its Supporting Information files.

## Abstract

Mammalian cells release different types of vesicles, collectively termed extracellular vesicles (EVs). EVs contain cellular microRNAs (miRNAs) with an apparent potential to deliver their miRNA cargo to recipient cells to affect the stability of individual mRNAs and the cells' transcriptome. The extent to which miRNAs are exported via the EV route and whether they contribute to cell-cell communication are controversial. To address these issues, we defined multiple properties of EVs and analyzed their capacity to deliver packaged miRNAs into target cells to exert biological functions. We applied well-defined approaches to produce and characterize purified EVs with or without specific viral miRNAs. We found that only a small fraction of EVs carried miRNAs. EVs readily bound to different target cell types, but EVs did not fuse detectably with cellular membranes to deliver their cargo. We engineered EVs to be fusogenic and document their capacity to deliver functional messenger RNAs. Engineered fusogenic EVs, however, did not detectably alter the functionality of cells exposed to miRNA-carrying EVs. These results suggest that EV-borne miRNAs do not act as effectors of cell-to-cell communication.

## Author summary

The majority of metazoan cells release vesicles of different types and origins, such as exosomes and microvesicles, now collectively termed extracellular vesicles (EVs). EVs have gained much attention because they contain microRNAs (miRNAs) and thus could

**Funding:** Funding for this research was provided by Deutsche Forschungsgemeinschaft grant nr. SFB1064/TP A13 to WH, Deutsche Forschungsgemeinschaft grant nr. SFB-TR36/TP A04 to WH, Deutsche Krebshilfe, grant nr. 70112875 to WH, and by the National Cancer Institute (grant nr. CA70723). The funders had no role in study design, data collection and analysis, decision to publish, or preparation of the manuscript.

**Competing interests:** The authors have declared that no competing interests exist.

regulate their specific mRNA targets in recipient or acceptor cells that take up EVs. Using a novel fusion assay with superior sensitivity and specificity, we revisited this claim but found no convincing evidence for an efficient functional uptake of EVs in many different cell lines and primary human blood cells. Even EVs engineered to fuse and deliver their miRNA cargo to recipient cells had no measurable effect on target mRNAs in very carefully controlled, quantitative experiments. Our negative results clearly indicate that EVs do not act as vehicles for miRNA-based cell-to-cell communication.

## Introduction

Cells release different types of extracellular vesicles (EVs) into the extracellular space. EVs have been reported to transfer proteins and RNA molecules from cell to cell and are thought to be important vehicles of intercellular communication [1]. They are released by a broad range of cell types and have been found in all body fluids, including blood, urine, and saliva [2–4]. A class of EVs, termed exosomes, can originate from cytoplasmic multivesicular bodies (MVB), which fuse with the cellular plasma membrane to release a burst of EVs. In addition, single EVs can also directly emerge from the plasma membrane to give rise to microvesicles [5,6]. Exosomes are 40–100 nm in diameter, and microvesicles can be up to 1000 nm. They have similar biophysical properties and are therefore difficult to study separately [7]. In this work, we use the term EVs to include both classes of vesicles.

microRNAs (miRNAs) are small noncoding RNAs 19–22 nt in length, which have important roles in the post-transcriptional regulation of gene expression. miRNAs act intracellularly, but a small fraction are found in the extracellular environment and in different biological fluids *in vivo* as well as in cell-culture media *in vitro* [8,9]. Extracellular miRNAs are thought to be promising circulating biomarkers for several cancers and other diseases [10] as the cancerous cells release typical miRNA species of diagnostic value [11–13].

miRNAs within EVs have been characterized extensively. EVs released from different cell types contain miRNAs and are delivered to other target cells, where the miRNAs regulate their cognate target genes at the posttranscriptional level [14–17]. miRNAs have been considered to be exclusively released within and protected by EVs since circulating miRNAs are extremely stable and resistant to RNases and have been detected in EV preparations purified from many cell types [18,19]. In contrast, two groups independently reported that extracellular miRNAs are rarely contained in EVs but predominantly associated with RNA binding proteins, such as AGO2, that protect extracellular miRNAs from degradation by ubiquitous RNases [10,20,21]. How these EV-free miRNAs are released from cells and whether they are taken up and functional in recipient cells is still uncertain. In addition, an analysis of the stoichiometry of miRNAs contained in exosomes suggested that EVs carry only low numbers of miRNA molecules that are too few to make a biologically significant difference in recipient cells [21]. A deeper knowledge of cell-to-cell transfer of miRNAs is needed to further address this controversy, but it is extremely challenging to characterize the functionality of EV-borne miRNAs in recipient cells because they usually express the very same endogenous miRNAs species. This major problem precludes an accurate evaluation of the transferred miRNAs and their functionality.

Here we used viral miRNAs released from human B cells latently infected with Epstein-Barr virus (EBV) as a model to characterize the role of EV-contained viral miRNAs and their known functions in target cells. In this model, viral miRNAs delivered by EVs are genetically distinct from human miRNAs. Thus, the transferred miRNAs, their uptake and functions in recipient cells can be easily discriminated from host miRNAs. This model has been already

employed by others to study the role of EBV's miRNAs. Several groups reported that EVs from latently EBV B cells can deliver viral miRNAs to target cells regulating certain cellular mRNA targets in monocyte-derived dendritic cells or the monocytic cell line THP-1 [14,15,22]. However, these studies did not distinguish between effects mediated by miRNAs and those mediated by EVs. Our approach makes use of engineered human B cells, which are infected with mutant EBVs that encode or are devoid of EBV's miRNAs [23] yet release EVs that contain the entire spectrum of cellular miRNAs, providing an important reagent and reference for experimental validation.

In our study, we confirmed that latently EBV-infected cells release human as well as viral miRNAs of which only a small fraction co-purified with EVs. Depending on the miRNA species, only 5–11% of all extracellular miRNAs were found inside EVs. Employing a very sensitive and novel assay, we observed a substantial and perhaps specific binding of EVs to a range of different target cells but failed to detect a fusion between EVs and recipient cells and thus a release of EV cargo into their cytoplasm. Given these findings, not surprisingly, we could not confirm a functional role of EV-borne miRNAs in recipient cells. Our experiments also revealed that single-molecule copies of three different viral miRNA species are found in 300 to $1.6 \times 10^4$ EVs, mainly depending on the miRNAs' abundance in EV-releasing cells. In summary, this work documents that an EV-mediated transfer of their miRNA cargo to all recipient cells tested is functionally irrelevant.

## Results

### Extracellular vesicles contain only a minority of extracellular miRNAs

We used human lymphoblastoid B-cell lines (LCLs) latently infected with EBV as a source of EVs to investigate their miRNA content and functionality. EBV encodes 44 miRNAs [23] with known or presumed targets and functions [24,25], leading to the release of EVs with cellular as well as viral miRNAs.

First, we validated our method to enrich and purify EVs. To do so, LCLs were cultured for 72 hours in cell-culture medium depleted of bovine EVs contained in fetal calf serum (S1 Fig), and cell-derived EVs were isolated from conditioned medium by several steps of differential centrifugation (Fig 1A). Two low-speed centrifugations removed cells and cellular debris (pre-purification), and then two steps of ultracentrifugation pelleted and concentrated EVs ('miniUC pellet' in Fig 1A). Resuspended EVs were further purified by floating in discontinuous iodixanol (Optiprep) gradients (Fig 1A). Finally, EVs were quantitated by nanoparticle tracking analysis (NTA), carefully validated for sensitivity and linearity of analysis (S2 Fig). EV-sized particles were found at the top of the gradient in fractions 2 and 3 at densities of around 1.05 g/mL (Fig 1B). The presence of EVs in these fractions was also confirmed by western blot immunodetection with antibodies directed against the human protein TSG101 and the viral protein LMP1 (Fig 1C), which are both enriched in EVs [26]. EVs in fractions 2 and 3 were free of other cellular organelles, such as endoplasmic reticulum (ER), as indicated by the absence of calnexin (Fig 1C). Representative members of cellular and viral miRNA were contained in fractions 2 and 3, indicating that highly enriched EVs and miRNAs co-purify in these gradients (Fig 1D). The integrity and the quality of EVs in these two fractions were assessed by electron microscopy (Fig 1E).

To determine if miRNAs that co-purify with EVs constitute the majority of the extracellular miRNAs released from cells, RNA was extracted from samples obtained from all steps of EV preparation before and after discontinuous flotation density gradient ultracentrifugation (Fig 2A). Characterization of the RNA molecules using an Agilent Bioanalyzer showed a progressive enrichment of small RNAs and a substantial loss of ribosomal RNAs in the more advanced

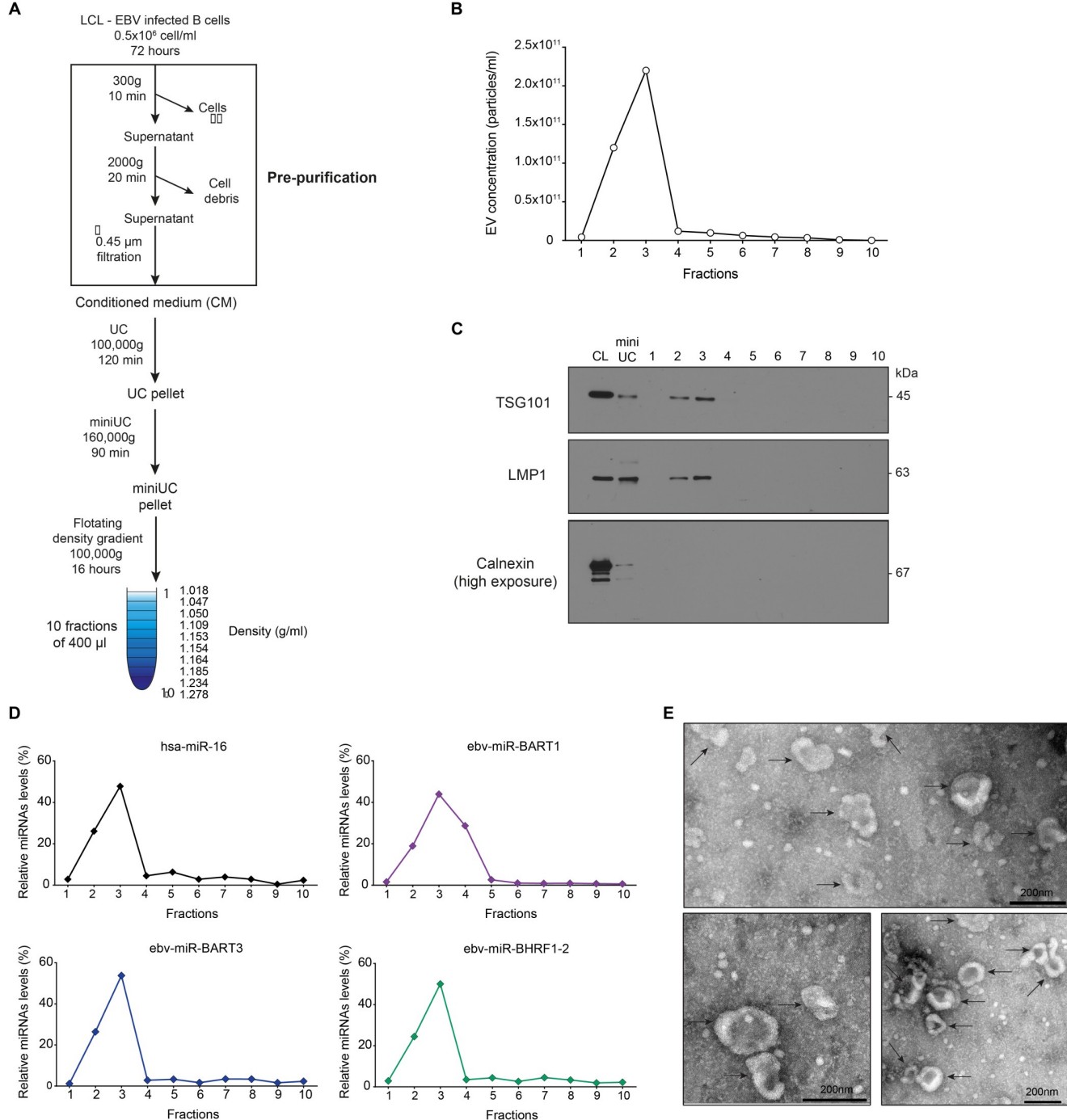

**Fig 1. Isolation and characterization of extracellular vesicles from EBV-infected cells. (A)** Schematic overview of our method of extracellular vesicles (EVs) enrichment and purification. EBV-infected B cells, lymphoblastoid cell lines (LCLs), were seeded at an initial density of $0.5 \times 10^6$ cells/ml in cell-culture medium containing 2% of EV-depleted fetal calf serum (see Materials and Methods) and processed as indicated. **(B)** The concentrations of EVs in the 10 fractions after iodixanol (Optiprep) floating density gradient centrifugation were measured by nanoparticle tracking analysis (NTA). One representative preparation is shown. **(C)** Western blot immunodetection of the EV marker protein TSG101 and the EBV protein LMP1, which are enriched in EVs, and Calnexin as a negative control in the 10 fractions. 5 µg of the cell lysate (CL) or resuspended 'miniUC pellet' (mini UC) preparations as indicated in panel A were used as controls. Per density gradient fraction 10, 20, or 60 µl were loaded onto the gels to detect LMP1, TSG101, or Calnexin, respectively. One representative preparation of three is shown. **(D)** Relative levels of four selected miRNAs (three viral miRNAs and a representative human miRNA) were analyzed by TaqMan RT-qPCR analysis to determine their physical density characteristics after floating density ultracentrifugation. All miRNAs are found in fractions 2–4. One representative quantification of three is shown. **(E)** Electron microscopic analysis of negative-stained EVs after iodixanol density gradient purification. Scale bars are 200 nm.

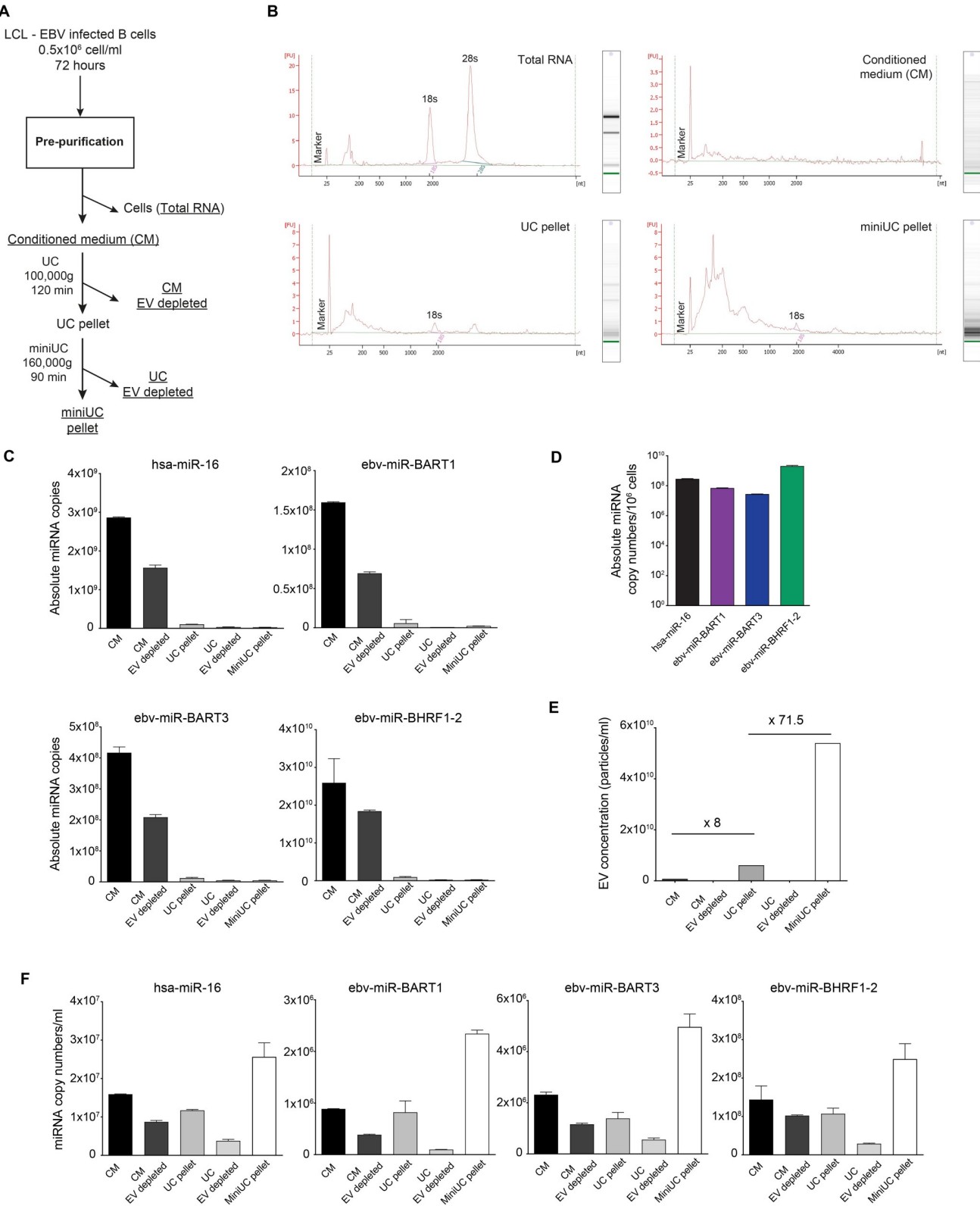

**Fig 2. microRNAs quantitation during different steps of EV purification. (A)** Schematic overview of EV sample collection and purification. After 72 h of LCL cell culture, we collected (i) the cell supernatant ('conditioned medium'; 180 ml), (ii) the pellet after ultracentrifugation ('UC pellet'; 9 ml), (iii) the supernatant after ultracentrifugation ('conditioned medium EV depleted' (180 ml), (iv) the pellet after the second ultracentrifugation step ('miniUC

pellet'; 1 ml), and (v) the remaining supernatant ('UC EV depleted'; 9 ml). RNA was extracted from 200 µl of each sample. As a control 'total RNA' from $1 \times 10^6$ LCLs was prepared. **(B)** Electropherograms after Bioanalyzer analysis of four samples are shown with their extinction profiles. **(C)** Using a Taqman stem loop RT-qPCR protocol, absolute copy numbers of four miRNAs were determined in different steps of EV purification as shown in panel A. For each of the four individual miRNA, a regression function with a synthetic miRNA oligonucleotide was generated by RT-qPCR for its absolute quantification. The different sample volumes were considered, according to the legend of panel A. **(D)** Absolute copy numbers of a human and three viral miRNAs in $10^6$ EBV-infected B cells (LCLs) are provided. **(E)** Concentrations of EV particles contained in different samples as indicated were measured by NTA. Numbers indicate fold-changes between 'conditioned medium' and 'UC pellet' (×8) and 'conditioned medium' and 'miniUC pellet' (×71.5) as explained in panel A. **(F)** Concentrations of four individual miRNAs (three viral miRNAs and a representative cellular miRNA) are shown in the different steps of EV purification as illustrated in panel A. Error bars in panels C–F indicate mean and SD of triplicates. Data obtained from one experiment of two independent experiments are shown.

steps of EV purification (Fig 2B). For example, the RNA profile contained in the 'miniUC pellet' indicated a clear enrichment of RNA molecules shorter than 200 nt on average and the depletion of RNAs with lengths corresponding to ribosomal RNAs (Fig 2B).

RNAs from different steps of purification were subjected to absolute miRNA quantification by TaqMan RT-qPCR analysis as described in Materials and Methods and specified in S3 Fig. Prior to RNA extraction and purification, samples were spiked with $10^7$ copies (unless stated otherwise) of a synthetic miRNA, cel-miR-39 (http://www.mirbase.org/), as an external standard and independent reference to account for variabilities during RNA purification and first-strand cDNA synthesis. Synthetic RNA oligonucleotides with the sequences of the mature miRNAs hsa-miR-16 (human), the viral miRNAs ebv-miR-BART1, ebv-miR-BART3, and ebv-miR-BHRF1-2, as well as the reference miRNA cel-miR-39, were used as standards for subsequent absolute miRNA quantification and data normalization.

After ultracentrifugation at 100,000 g for 120 min, the majority of miRNAs did not sediment but remained in the conditioned medium (CM in Fig 2C), suggesting that few miRNAs are associated with EVs as reported [10,20]. The concentration of the four viral miRNAs in conditioned medium correlated approximately with their intracellular abundance (Fig 2C and 2D). The concentration of EV particles as quantified by NTA increased eightfold and more than 70-fold in the 'UC pellet' and the 'miniUC pellet' preparations, respectively, compared with EV concentrations in 'conditioned medium' (Fig 2E), but most of the miRNA molecules did not co-purify with EVs. Among the four different miRNAs, only a modest enrichment in the order of 1.6 to 2.7-fold was observed, comparing 'conditioned medium' and 'miniUC pellet' preparations (Fig 2F), which led us to conclude that only a minute fraction of miRNAs is associated with EVs.

To validate this hypothesis, we used size exclusion chromatography (SEC) as a second alternative method of EV isolation (Fig 3A), which allows separating EVs (Fig 3B; fractions 7–9) from free protein (fractions 14–20) as documented by NTA, protein quantification (Fig 3B), and western blot immunodetection of the viral protein LMP1 (Fig 3C). We quantified the miRNA levels in each of the 20 fractions after SEC. In agreement with our initial findings, only low levels of miRNAs were detected with EVs in fractions 7–9, whereas the majority of miRNAs in fractions 16–18 (Fig 3D) co-purified with free protein (Fig 3B and 3D).

These results documented that only a minor fraction of extracellular miRNAs of both EBV and human origin is associated with EVs.

## Binding of extracellular vesicles to target cells suggests specific interactions

Several groups reported that EVs containing EBV miRNAs are taken up from recipient cells where viral miRNAs regulate the expression of their cognate cellular transcript targets [14,15]. To study the functional role of EV-delivered viral miRNAs in target cells, we engineered pairs of LCLs from the same B-cell donors that differ only in the presence (or absence) of viral miRNAs in EVs. Using these pairs of LCLs, we characterized the functions of viral miRNAs

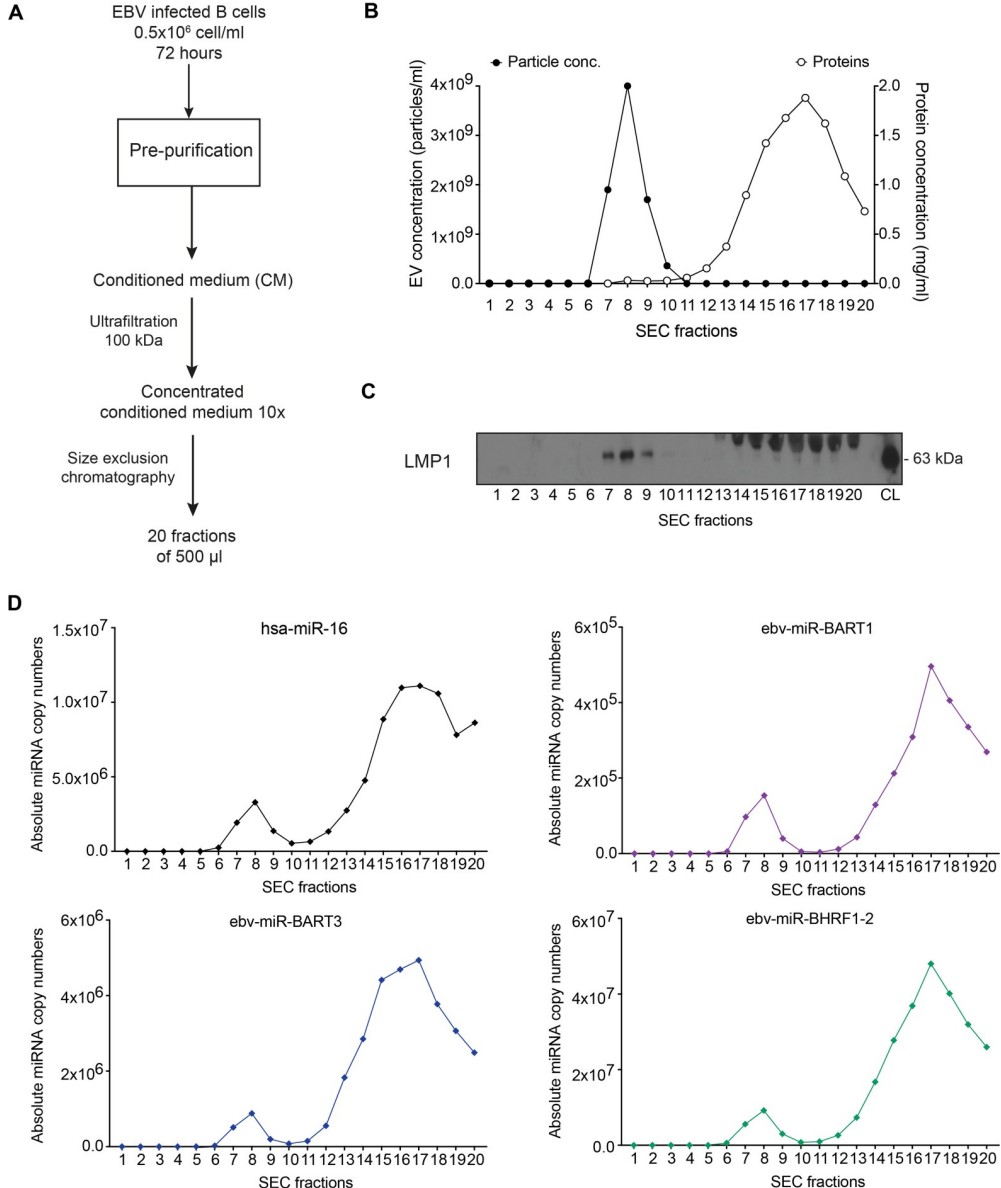

**Fig 3. The majority of miRNAs does not co-purify with extracellular vesicles. (A)** Schematic overview of the purification of EVs by size-exclusion chromatography (SEC). EBV-infected B cells (LCLs) were seeded at an initial density of $0.5\times10^6$ cells/ml in medium containing 2% of EV-free FCS (see Materials and Methods). After 72 h, the supernatant was harvested and different steps of centrifugation were used to remove cells (300 g for 10 min) and cell debris (2000 g for 20 min), followed by filtration through a 0.45-μm mesh size filter. The conditioned cell-culture medium was further concentrated 10-fold to a final volume of 1 ml using a 100-kDa centrifugal ultrafiltration device (Amicon). The concentrated conditioned medium was then loaded onto a size-exclusion chromatography qEV column (Izon Science Ltd). 20 fractions of 500 μl each were collected. **(B)** Concentration of EV particles and protein was measured in each fraction by NTA and a colorimetric Bradford assay, respectively. **(C)** EVs were found in fractions 7, 8, and 9 as confirmed by Western blot immunodetection with an LMP1-specific antibody. LCL cell lysate (CL) was used as positive control. The viral LMP1 protein is highly enriched in the membranes of EVs. A non-specific band of about 70 kDa observed in EV-free, protein-enriched fractions 14–20 likely stems from immunoglobulin heavy chain molecules produced by EBV-infected B cells. **(D)** Using a TaqMan stem loop RT-qPCR, absolute copy numbers of miRNAs were determined in each fraction after size-exclusion chromatography. For each miRNA, a standard regression obtained with a corresponding synthetic miRNA oligonucleotide was used for absolute miRNA quantification. The results show one representative experiment of three.

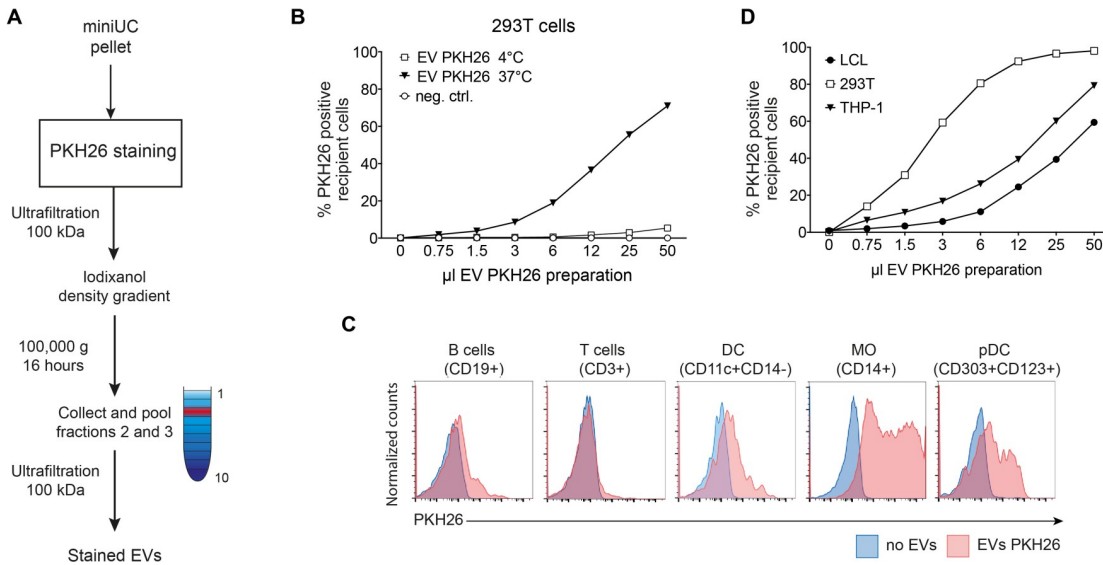

**Fig 4. EVs from EBV-infected B cells interact preferentially with certain cell types. (A)** Schematic overview of the labeling and purification of EVs, which were isolated as described in Fig 1A. Resuspended EVs contained in the 'miniUC pellet' were stained with the dye PKH26 as described in Materials and Methods. After staining, the EVs were concentrated using a 100-kDa Amicon centrifugal filter and then loaded at the bottom of an iodixanol (Optiprep) floating density gradient. After 16 h at 100,000 g, fractions 2 and 3 containing the EVs were pooled and washed three times with PBS using a 100-kDa Amicon centrifugal filter unit to remove unbound dye. Concentrated and PHK26-stained EVs were finally resuspended and used in the assays shown in panels B and C. A negative control without EVs was mock stained with the same amount of dye and purified in parallel. **(B)** 293T cells were incubated with increasing volumes of PKH26-stained EVs or the negative control (neg. ctrl.) at 37˚C. As control, 293T cells were incubated with the same volume of stained EVs at 4˚C. After 4 h, 293T cells were analyzed by flow cytometry. The percentage of PKH26 positive cells is shown as a function of EV volume dose. **(C)** Peripheral blood mononuclear cells (PBMCs) were incubated with 100 μl of PHK26-stained EVs (about 1,000 EVs per cell). After 4 h, PBMCs were stained with antibodies specific for different cell types and analyzed by flow cytometry as indicated above each histogram. A representative example of two independent experiments is shown for each panel. **(D)** EBV-infected B cells (LCL), THP-1, or 293T cells were incubated as described in panel B at 37˚C for 4 h and analyzed for the fraction of PKH26-positive cells. The results show one representative experiment of three.

that purify with EVs to determine the number of EVs needed to deliver a functional dose of EBV miRNAs to recipient cells.

We sought to identify biologically relevant target cells of EVs released from LCLs. To do so, highly enriched EVs (miniUC pellet) were stained with PKH26, a red fluorescent lipid dye, and they were purified by discontinuous floating density gradient centrifugation to remove unbound, free dye from the EV preparation (Fig 4A). After purification, we incubated 293T cells with different amounts of PKH26-labeled EVs and corresponding volumes of negative control (PKH26 dye only, purified in parallel). After 4 hours at 37˚C, 293T cells showed a dose-dependent increase of fluorescent, PKH26-positive cells as quantitated by flow cytometry (Fig 4B), which was not observed when the cells were incubated with the negative control (ctrl, Fig 4B) or with PKH26 stained EVs at 4˚C (Fig 4B). We incubated the PKH26-labeled EVs with human PBMCs at 37˚C for 4 hours and analyzed selected cellular subpopulations by flow cytometry. As reported, monocytes and plasmacytoid DC (pDCs) were intensely stained, whereas human B lymphocytes and dendritic cells showed a lower level of PKH26 staining (Fig 4C) [27]. Interestingly, T cells did not show any staining with PKH26 even when high numbers of EVs were used, suggesting a specific and selective binding of LCL-derived EVs to certain primary cell types and established cell lines (Fig 4D).

Importantly, PKH26 staining of cells after incubation with dye-labeled EVs indicates a robust cellular interaction and probably also an enrichment of EVs at the level of the cells' plasma membranes. As this method cannot distinguish among binding, internalization, or

delivery of EVs and their cargo, we developed a novel functional assay to detect and quantitate EV fusion with or uptake by their putative target cells.

## EV-borne delivery of cargo to target cells is inefficient

To address this important issue, we developed a simple and rapid assay to determine if EVs deliver their content to recipient cells and to quantitate the efficiency of this process. The assay is based on a β-lactamase reporter (BlaM) molecule and a FRET (Fluorescence or Förster Resonance Energy Transfer) -coupled substrate that can be analyzed by flow cytometry and fluorescence microscopy. A lipophilic, esterified form of the substrate (CCF4-AM) can be easily loaded into the cytoplasm of any cell, where it is rapidly converted into its negatively charged form, CCF4, which is retained in the cytosol. CCF4 is very stable, and its ß-lactam ring is only cleaved when β-lactamase is delivered intact to the cells. The non-cleaved CCF4 substrate and its cleaved derivative can be easily differentiated and quantified by flow cytometry. The BlaM assay has been used extensively with HIV particles to analyze their fusion with and entry into different target cell populations [28,29].

To study the functions of EVs, we fused a synthetic, codon-optimized version of the BlaM gene to the carboxy-terminus of CD63, a member of the tetraspanin family and cellular receptor enriched in EVs ([30,31] and references therein). We expressed the CD63-β-lactamase protein (CD63-BlaM) transiently in 293T cells or constitutively in 293T cells using a lentiviral vector. EVs harvested from the supernatants of donor cells carried the CD63-BlaM protein, including its intact β-lactamase activity (Fig 5A).

As a positive control, we collected EV-containing supernatant from 293T transiently co-transfected with two expression plasmids encoding CD63-BlaM and the vesicular stomatitis virus G (VSV-G). VSV-G is broadly used to pseudo-type retroviral or lentiviral gene vectors because the glycoprotein is incorporated into viral envelopes and ensures a broad tropism and high transduction efficacy of these viral vectors. VSV-G is also incorporated into the membranes of EVs, where it confers membrane fusion with other cells ([32] and references therein).

After isolating EVs from CD63-BlaM 293T cells, we incubated recipient 293T cells with EVs for 4 hours. Then the cells were washed, loaded with CCF4-AM, and analyzed by flow cytometry. The CCF4 substrate was readily cleaved in cells incubated with EVs containing CD63-BlaM and VSV-G (Fig 5B, right panel), but no CCF-4 cleavage was detected when cells were incubated with CD63-BlaM-assembled EVs lacking VSV-G or when control cells without EV treatment were analyzed (Fig 5B, middle and left panels). This experiment indicates that EVs deliver β-lactamase with high efficiency to target cells, in principle, but only when pseudo-typed with VSV-G.

We also purified CD63-BlaM-assembled EVs with or without VSV-G and stained both preparations with the dye PKH26 as in Fig 4 to determine if EV surface binding and ß-lactamase activity correlate. EVs from both preparations were purified on iodixanol gradients and eight fractions were harvested and analyzed (Fig 5C). This experiment was designed to be able to identify also infrequent fusion events of unmodified EVs, i.e., those without VSV-G, that we failed to observe in Fig 5B. For this reason, higher doses of EVs were used using fixed volume aliquots from all fractions. As a consequence, we detect binding of PKH26 stained EVs from fractions 2 to 6 (Fig 5C, left panel), although, as shown in Fig 1B, fractions 4 to 6 contained EVs at much lower concentrations. Due to the high EV doses used in these experiments, we obtained very high levels (up to 100%) of PHK26 and BlaM-positive cells in fractions 2 to 4. Both EV preparations stained 293T target cells similarly (Fig 5C, left panel), but only VSV-G-assembled EVs induced cleavage of CCF4 (Fig 5C, right panel), documenting that EVs

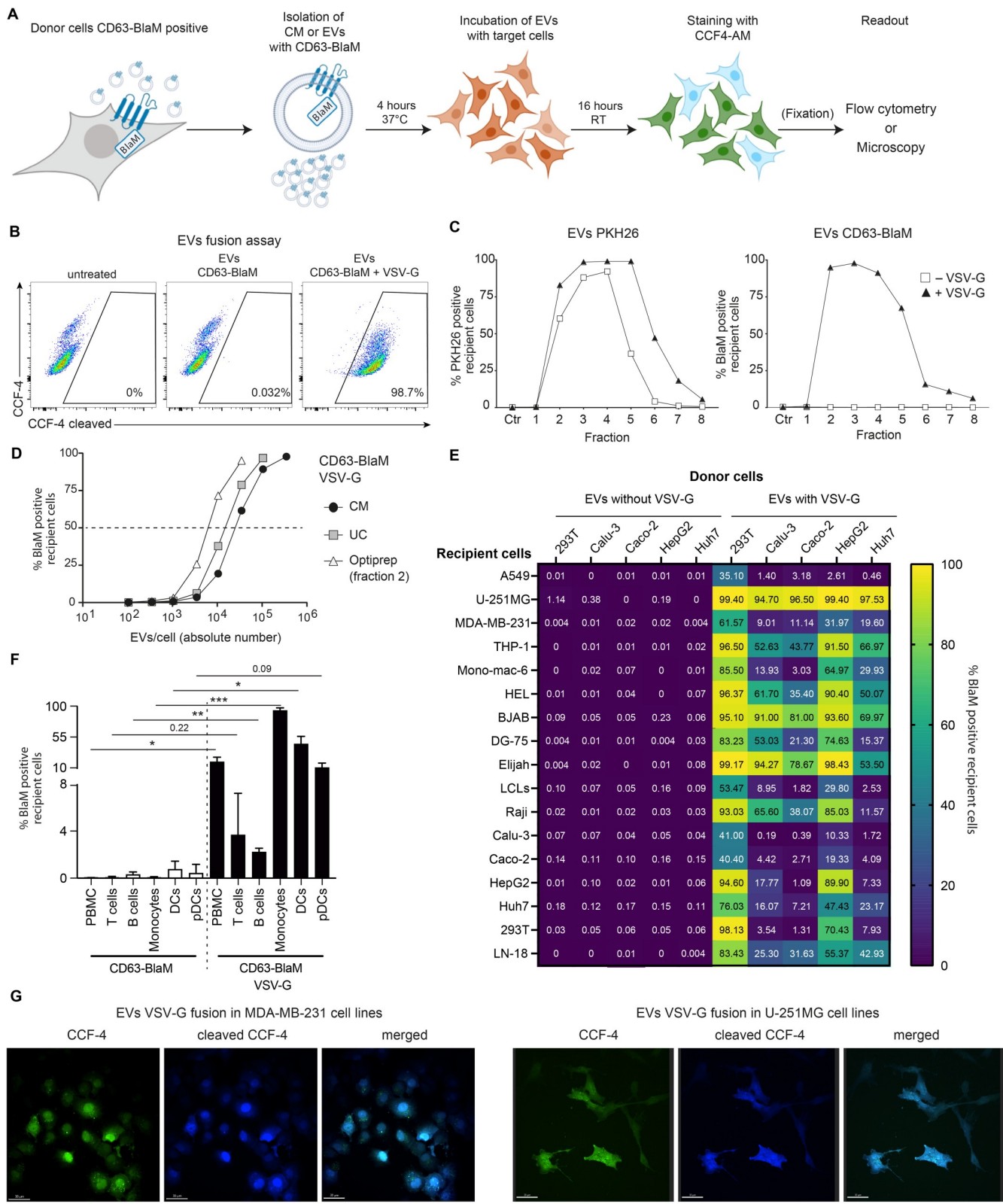

**Fig 5. EVs do not deliver their cargo to recipient cells unless EVs carry a fusogenic glycoprotein.** The CD63-conjugated β-lactamase (CD63-BlaM) fusion protein serves as a reporter to investigate the uptake of EVs by recipient cells. **(A)** The flow chart depicts the fusion assay starting with conditioned medium (supernatant) or purified preparations of CD63-BlaM containing EVs from donor cells that express the ß-lactamase fusion protein. The target cells were

incubated for 4 h, loaded with CCF4-AM substrate overnight, fixed depending on the cell type, and analyzed by flow cytometry. Fixation (in parenthesis) is an optional step and not essential for the assay to work. (The graphics was created with BioRender.com.) **(B)** $2 \times 10^5$ 293T recipient cells were incubated for 4 hours with 50 μl of concentrated EVs (about 50,000 EVs per cell) isolated from 293T donor cells transiently transfected with two expression plasmids encoding CD63-BlaM and VSV-G or with a single plasmid encoding CD63-BlaM, only. After loading the CCF4-AM substrate, 293T recipient cells were analyzed by flow cytometry. CCF4 was readily cleaved in cells incubated with CD63-BlaM assembled EVs equipped with VSV-G (right panel), but not in CCF4-loaded cells that had been incubated with EVs assembled with CD63-BlaM, only (middle panel) or untreated cells (left panel). **(C)** 293T donor cells were transfected with an expression plasmid encoding CD63-BlaM alone or together with a second plasmid encoding VSV-G as in panel B. Both EV-containing supernatants were harvested, purified, and the 'UC pellet' was stained with PKH26 as in Fig 4A. Identical volume samples (50 μl) of the eight fractions obtained after iodixanol (Optiprep) density gradient centrifugation were incubated with $2 \times 10^5$ 293T recipient cells and analyzed for PKH26 fluorescence and CCF4 cleavage by flow cytometry. Both types of EVs contained in fractions 2–5 of the gradients bound equally well to 293T cells (left panel) as indicated by the percentage of PKH26-positive cells. BlaM-positive cells were only detected when incubated with VSV-G assembled EVs contained in fractions 2–5 but not in cells incubated with VSV-G negative EVs (right panel). **(D)** 293T cells were transiently transfected with two expression plasmids encoding CD63-BlaM and VSV-G and the conditioned medium (CM) was collected. It was concentrated (UC pellet as in Fig 4A) and purified by iodixanol (Optiprep) density gradient centrifugation. The concentrations of EVs in conditioned medium (CM), the resuspended UC pellet and fraction 2 of the gradient were analyzed by NTA. $2 \times 10^5$ 293T target cells were incubated for 4 h with increasing amounts of the three EV preparations and the percentage of BlaM-positive cells was determined and plotted versus the number of EV particles used per cell. **(E)** Heatmap of a set of EV fusion assays with five donor cells and 17 different recipient cells. 293T, Calu-3, Caco-2, HepG2 and Huh7 were engineered to express CD63-BlaM stably after lentiviral transduction. Where indicated, the cells were transiently transfected with an expression plasmid encoding VSV-G. 50 μl of EVs from donor cells were purified and incubated with $2 \times 10^5$ recipient cells from 17 different cell lines for 4 h. The cells were loaded with CCF4-AM substrate, fixed and analyzed by flow cytometry. Means of three technical replicates of the transfer experiments are shown. All single datasets are shown in S4A Fig. The concentration of EVs used is shown in S4C Fig. **(F)** PBMCs were incubated with EVs (about 35,000 EVs per cell) obtained from conditioned medium of 293T cells transiently transfected with the CD63-BlaM encoding plasmid, only, or together with an expression plasmid encoding VSV-G. PBMCs were incubated for 4 h, loaded with CCF4-AM substrate, and stained with antibodies to distinguish different cell types contained in PBMCs. The cells were analyzed by flow cytometry and gated for their identity (T cells, B cells, monocytes, DCs, pDCs) and percentages of CCF4 cleavage. Mean and SD of three independent donors are shown. Asterisks indicate statistical significance by paired two-tailed t test. ($^*P \leq 0.05$; $^{**}P \leq 0.01$; $^{***}P \leq 0.001$). **(G)** U-251MG and MDA-MB-231 cells were used as recipient cells and incubated with EVs from 293T cells transfected with plasmids encoding CD63-BlaM and VSV-G as in panel E. The cells were analyzed by confocal microscopy. Scale bars is 30 μm. Controls are provided in S5B Fig.

efficiently bind to the surface of target cells but do not deliver their cargo into the cytoplasm unless they are equipped with an ectopic fusogenic moiety, such as VSV-G. Our findings suggest that fusion of EVs with cellular membranes of recipient cells with or without receptor-mediated uptake and endocytosis is extremely inefficient (Fig 5C).

In addition, we assessed the physical number of EV particles by NTA (S2 Fig) to titrate purified VSV-G-pseudo-typed and CD63-BlaM assembled EVs using 293T as target cells (Fig 5D). Within 4 hours, about $1 \times 10^4$ physical EV particles per cell were sufficient to transduce half of the cells in this test, indicating a good efficacy and high sensitivity of the fusion assay.

To further validate our findings, other donor and recipient cell combinations were tested for EV-mediated delivery of CD63-BlaM without VSV-G. Since only donor cells need to express the CD63-BlaM reporter to be incorporated into EV membranes, any cell can be used as potential recipient in our EV fusion assay. We engineered 293T, Calu-3, Caco-2, HepG2 and Huh7 cells to express CD63-BlaM constitutively and at high levels using lentiviral transduction techniques. EVs purified from these five different cell lines were incubated with 17 different recipient cells. CD63-BlaM assembled EVs lacking VSV-G showed no measurable or extremely low (U-251MG cells) EV uptake (Fig 5E) and did not spontaneously deliver their cargo to most of the different cell types in PBMCs from several donors (Fig 5F). Only upon transient co-transfection of 293T cells with VSV-G, EVs contained in the supernatant of these cells readily released their content into recipient cells with different efficacies. Interestingly, recipient cells did not necessarily show the highest uptake when tested with their cognate donor cells (Figs 5E and S4). Similar results were obtained using LCLs as donor cells (S5A Fig).

Next, we tested whether cleavage of the CCF4 substrate could also be detected by fluorescence microscopy. No blue cells were detected in untreated control cells (S5B Fig) but cleaved CCF4 was readily detected in cells incubated with CD63-BlaM containing EVs assembled with VSV-G (Fig 5G). The blue signal deriving from the CCF4 cleaved product was homogeneously distributed in the cells' cytoplasm suggesting a direct delivery of CD63-BlaM into this cellular

compartment. These observations show that inefficient delivery and transfer of functional cargo contained in the lumen or in the membranes of EVs from six different cell types, including 293T and LCLs is due to the lack of an EV-intrinsic fusogenic activity, which nevertheless can be easily introduced by expressing VSV-G during EV biogenesis.

## EV-borne miRNAs do not regulate their cognate 3′-untranslated region targets in sensitive reporter assays

Our previous experiments failed to show a convincing fusion of EVs with membranes of potential recipient cells, but the experiments did not directly determine if miRNAs might still be functionally transferred to target cells by alternative means. To assess this possibility, we used a dual luciferase reporter assay based on psiCHECK2 reporter plasmids equipped with two luciferases, Renilla and firefly. We introduced three copies of perfectly complementary target sites of three different viral miRNAs (ebv-miR-BART1, ebv-miR-BART3, or ebv-miR-BHRF1-2) into the 3′ untranslated region (UTR) of the Renilla luciferase reporter gene (Fig 6A). The signal from the firefly luciferase gene was used as an internal control for normalization.

First, we tested the sensitivity of this system by co-transfecting only 30 ng of the reporter plasmids with decreasing amounts of plasmid DNAs expressing the three single miRNAs of interest (Figs 6A and S6B). In general, the miRNA reporter system revealed a very high sensitivity. Transfection of as little as 1.56 ng DNA of a plasmid vector encoding ebv-miR-BHRF1-2 reduced the luciferase activity expressed from the corresponding reporter plasmid by half (Fig 6A). Fourfold more plasmid DNA was needed with two other expression plasmids encoding ebv-miR-BART1 or ebv-miR-BART3 to reach a similar level of repression (Figs 6A and S6B). These experiments also suggested that, depending on individual reporter plasmids, 20–300 miRNA copies per cell reduced the luciferase activity by half (S8 Fig).

Next, we employed 293T cells transiently transfected with 10 ng the three individual miRNA reporter plasmids each and added calibrated, increasing doses of purified EVs ('miniUC pellet') harvested from the CM of LCLs as the source of viral miRNAs. As a negative control, we used identically prepared EVs but purified from LCLs infected with a mutant EBV incapable of expressing viral miRNAs (ΔmiRNA EBV). We incubated the 293T reporter cells with up to $1x10^5$ EVs per cell for 24 hours but did not observe a specific reduction of Renilla luciferase activity (S6C and S7A Figs). (For comparison, $10^4$ VSV-G assembled EVs per cell were sufficient to transduce about 50% of all cells in Fig 5D.) A very similar result was obtained with THP-1 cells, which modestly bind EVs (Fig 4D). In this experiment, we used the maximum amount of EVs that the cells tolerated. A higher ratio of EVs per cell led to a reduction of the Renilla luciferase signal probably because a very high EV concentration was toxic to the cells (S7B Fig) as reported [33,34]. In fact, reduction of the Renilla signal was independent of the EV miRNA content (S7A Fig).

## Engineered EVs with reporter mRNAs deliver their cargo when assembled with VSV-G, but EV-delivered miRNAs are non-functional in recipient cells

The experiments so far suggested that either miRNAs levels in EVs are insufficient to regulate their target mRNAs in recipient cells upon EV-mediated delivery or that EV-contained RNA molecules *per se* are not functional in recipient cells. We addressed this fundamental uncertainty in two experimental settings, shown in Fig 6B and 6C. In these experiments, 293T cells were used both as donor cells for the generation of EVs and as recipient cells to perform functional analyses upon EV delivery. In the first approach (Fig 6B), donor cells were transfected

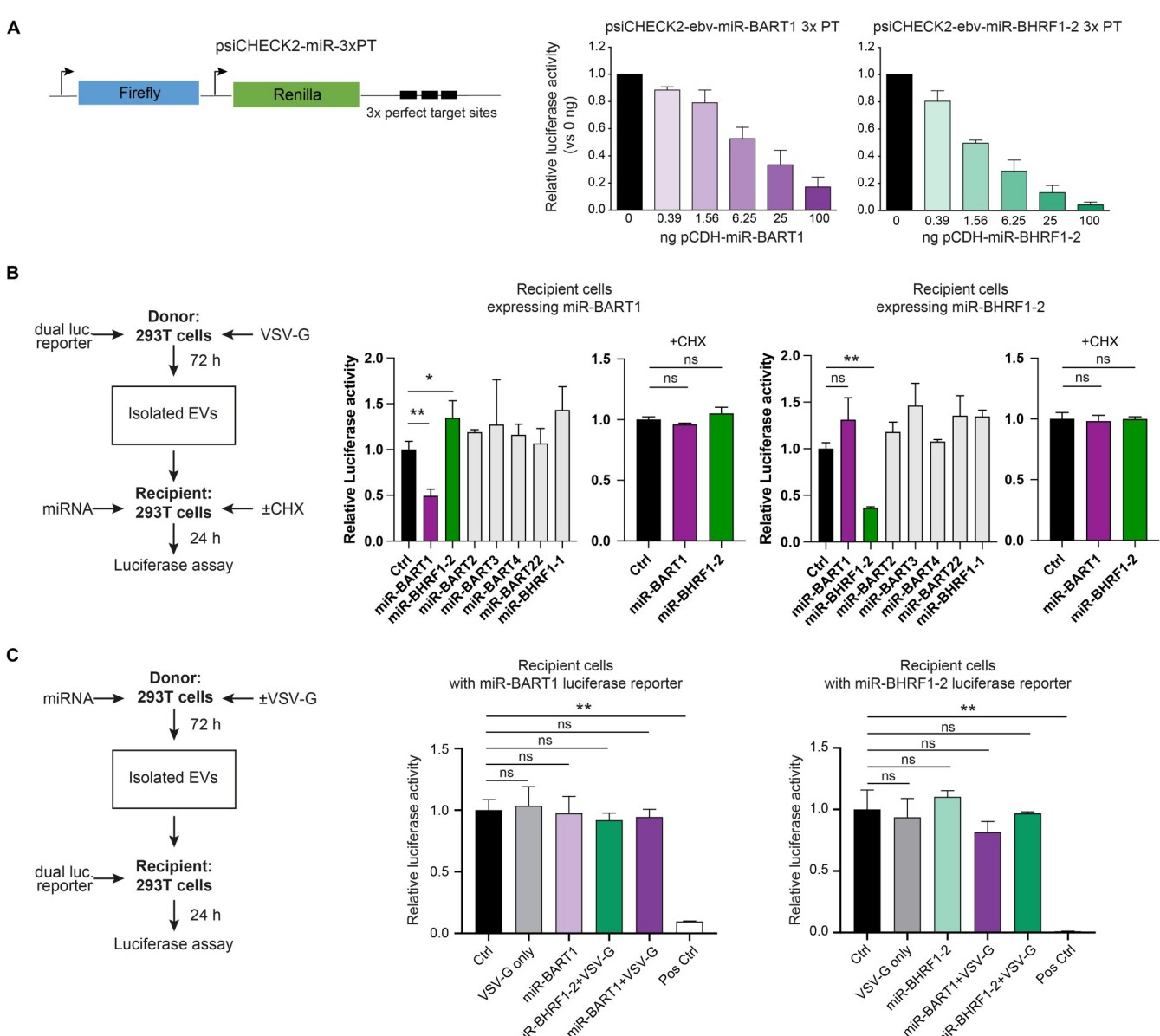

**Fig 6. Dual luciferase reporter assays indicate a functional EV-mediated transfer of mRNA transcripts but fail to detect miRNAs-dependent regulation of reporter transcripts in recipient cells. (A)** The design of the modified dual luciferase reporter plasmid, based on psiCHECK2, is shown, which encompasses the internal control firefly luciferase (used for normalization) and the reporter Renilla luciferase with three tandem copies of perfect complementary target sites (3xPT) of the miRNAs of interest inserted in the 3′UTR of the Renilla mRNA. 293T cells were transfected with 30 ng of the miRNA reporter plasmid containing 3xPT with increasing amounts of the corresponding miRNA expression vector (pCDH) starting with 390 pg up to 100 ng. At 24 h after transfection cells were lysed to determine the Renilla and firefly luciferase activities. Mean of three replicates is shown. **(B)** Reciprocal dual luciferase assays with EVs engineered to transfer luciferase-encoding mRNAs to recipient 293T cells expressing viral miRNAs. Left panel: overview of the principal components of the dual luciferase assay. 293T donor cells seeded in a 13-cm dish were transiently transfected with 12 μg of 3x PT psiCHECK2 dual luciferase reporter plasmid DNAs (shown in panel A) together with 8 μg of a VSV-G expression plasmid. 50 μl of purified EVs ($3.1x10^4$ particles/cell) were transferred to recipient 293T cells in 24-well plates transiently transfected with 500 ng expression plasmid DNA encoding miR-BHRF1-2 or miR-BART1 as indicated. As control, recipient 293T cells were also transfected with 500 ng of expression plasmid DNAs encoding miR-BART2, miR-BART3, miR-BART4, miR-BART22 or miR-BHRF1-1. As another control, the recipient cells were incubated with cycloheximide (CHX; 20 μg/ml) to abrogate translation. Middle and right panels: results of reporter assays lysates from recipient 293T cells expressing miR-BHRF1-2 or miR-BART1 as indicated. The transduced mRNAs encoding Renilla luciferase with the perfect complementary target sites (3xPT) for miR-BHRF1-2 or miR-BART1 (and firefly luciferase used for normalization) are translated and expressed in the recipient cells, but repressed in cells that contain the matching miRNA. Mean and SD of three independent donors are shown. Asterisks indicate statistical significance by paired two-tailed t test. ($^*P \leq 0.05$; $^{**}P \leq 0.01$; $^{***}P \leq 0.001$). **(C)** Dual luciferase assays with reporter constructs (3x PT psiCHECK2) shown in panel A were performed to investigate the functional transfer of EV-borne viral miRNAs to 293T recipient cells. Left panel: overview of the dual luciferase assay. 293T cells seeded in a 13-cm dish were transiently transfected with expression plasmids (12 μg) coding for miR-BHRF1-2 or miR-BART1 alone or in combination with an expression plasmid coding for VSV-G (8 μg). Medium was replaced with fresh medium

after 24 hours and the cells were incubated for another 72 h prior to harvest. 50 μl of EVs were transferred to 293T cells in a 24-wells plate transiently transfected with 10 ng of dual reporter 3x PT psiCHECK2 plasmid DNA as indicated. Middle and right panels: results of reporter assays show no repression of the luciferase reporters. Ctrl: conditioned medium from 293T donor cells transiently transfected with an expression plasmid encoding no miRNA; VSV-G: conditioned medium from 293T donor cells transiently transfected with a VSV-G encoding expression plasmid, only; Pos Ctrl: recipient 293T cells transiently co-transfected with both the miRNA expression plasmid and the corresponding 3x PT psiCHECK2 reporter plasmid. Mean and SD of three independent donors are shown. Asterisks indicate statistical significance by paired two-tailed t test. (*P ≤ 0.05; **P ≤ 0.01; ***P ≤ 0.001).

with plasmids encoding VSV-G and a dual luciferase reporter as shown in Fig 6A. The intention of this unusual experiment was to generate mRNAs for their EV-mediated delivery and their subsequent evaluation in suitable recipient cells.

VSV-G assembled EVs potentially bearing the mRNAs of the luciferase reporter system were purified. Before incubating the recipient cells, they were transfected with plasmids expressing either ebv-miR-BHRF1-2 or ebv-miR-BART1 to monitor the regulation of transduced reporter mRNAs upon their VSV-G-mediated EV delivery. EVs assembled with VSV-G successfully delivered functional mRNA transcripts encoding the luciferase reporter enzymes to recipient cells (Fig 6B). Treatment of recipient cells with cycloheximide abrogated miRNA-mediated repression of luciferase activity, showing that the assay discriminates between *de novo* translated Renilla and firefly luciferases and delivery of luciferase as active enzymes by EV-mediated transfer of proteins. These data demonstrate that the content of EVs is potentially functional when it is delivered into the cytoplasm of recipient cells but such delivery is rare. Based on similar experiments [35], we have estimated that the dose of EVs assembled with VSV-G lead to fewer than one hundred mRNA molecules encoding the luciferase reporter enzyme being delivered to the recipient cells. These estimates are necessarily imperfect; they reflect the uncertainties in the dose, the content of RNA in the EVs, the fraction of the EVs carrying VSV-G, and the rate of uptake of the EVs. Some of these parameters such as the fraction of VSV-G positive EVs or the rate of EV uptake are difficult to assess experimentally. It should also be noted that these experiments were not informative with EVs that had been assembled without VSV-G due to the extremely low signal levels in these trials that were close to background noise of the recording instrument (Fig 6B).

In a reciprocal approach (Fig 6C), donor cells were transfected with plasmids expressing miR-BHRF1-2 or miR-BART1 in combination with VSV-G. EVs generated from the supernatant of transfected cells were purified as before and incubated with recipient cells transfected with very low amounts of the dual luciferase reporter plasmids to monitor the functions of the EV-delivered miR-BHRF1-2 or miR-BART1 miRNAs. In these settings, EVs containing miR-BHRF1-2 or miR-BART1 failed to regulate Renilla luciferase activity in recipient cells, although the EVs had been assembled with VSV-G and the luciferase reporter was expressed at very low levels in the recipient cells, only.

These experiments demonstrate that EVs deliver RNA molecules such as mRNAs, in principle, that are then translated to functional protein in recipient cells, but EV-borne transfer of miRNAs is below the detection limit of this assay even when EVs are assembled with VSV-G.

## A single miRNA molecule co-purifies with hundreds of EVs

Since we observed no functional transfer of viral miRNAs even using VSV-G assembled EVs, we decided to calculate the average number of miRNA molecules associated with or contained within a single EV. We isolated and purified EVs from the supernatants of LCLs obtained from different donors from three different sources: the 'miniUC pellet' after two steps of ultracentrifugation (Fig 2A), the combined fractions 2 and 3 of the discontinuous iodixanol (Optiprep) floating density gradient (Fig 1A), and the combined fractions 7 and 8 after SEC (Fig 3D).

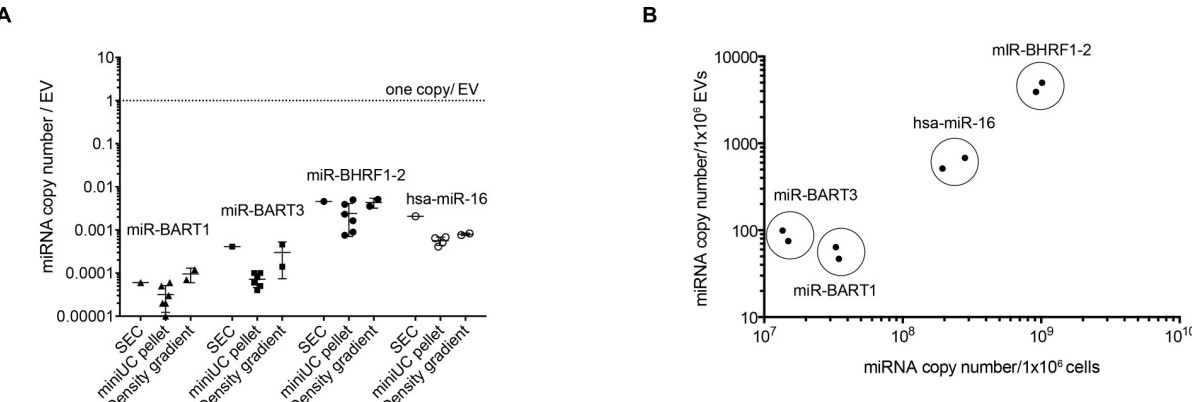

**Fig 7. Single miRNAs are very scarce in EVs.** Three different preparations of EVs were obtained from conditioned medium of LCLs infected with wild-type EBV. EVs were purified by differential centrifugation ('miniUC pellet'; Fig 2A), by size exclusion chromatography (SEC, fractions 7–9 were combined and analyzed; Fig 3A), or by iodixanol (Optiprep) density gradient centrifugation (fractions 2 and 3 were combined and analyzed; Fig 1D). Concentrations of EVs were determined by NTA. RNA was isolated from the three preparations (resuspended miniUC pellet, SEC and density gradient fractions), and absolute copy numbers of four different miRNAs were determined using a TaqMan stem loop RT-qPCR. For each miRNA, a standard curve of a synthetic RNA oligonucleotide (mimicking the miRNA) was established to determine absolute miRNA copy numbers in the preparations. **(A)** The graph shows the miRNA copy number per EV of three viral miRNAs and the human miRNA hsa-miR-16. **(B)** The graph indicates the miRNA copy numbers contained in $10^6$ EVs (y-axis) as measured by RT-qPCR and NTA, respectively, versus the miRNA copy number contained in $10^6$ cells (x-axis).

Absolute quantification of the different samples revealed that a single specific mature miRNA is found in 300 EVs (in case of the abundant ebv-miR-BHRF1-2) but more often in thousands of EVs (in case of the least abundant ebv-miR-BART1) (Fig 7). In three differently prepared EV batches, a single EV contained 0.00006±0.000037, 0.00018±0.00017 or 0.0034 ±0.0018 copies of ebv-miR-BART1, ebv-miR-BART3 or ebv-miR-BHRF1-2 molecules, respectively. Similar levels were observed with the human miR-16 miRNA (0.00084±0.00055 molecules per EV). Our data are in line with those of Chevillet and colleagues, who quantified human miRNAs contained in highly purified exosomes isolated from conditioned cell culture media or from plasma samples obtained from healthy individuals or patients with ovarian and prostate cancers [21].

In summary, our results demonstrate that the majority of extracellular miRNAs from cell-culture supernatants are not associated with or contained within EVs. The very low copy number of miRNAs found in highly purified EV preparations, together with the very inefficient uptake of EVs, strongly argue against a role of EVs in transferring functionally relevant amounts of miRNAs to recipient cells to modulate their transcriptome or gene expression profiles.

## Discussion

Many groups have reported that miRNAs contained in EVs are involved in a wide range of paracrine and endocrine biological activities and fulfil important functions in different types of target cells [16,17,36–39] (for a small selection of papers). Similarly, viral miRNAs encoded by EBV were reported to be delivered via exosomal transfer from latently infected B cells to recipient cells where viral miRNAs repress their cognate cellular mRNA targets [14].

How cellular or viral miRNAs make their way into extracellular vesicles has been elusive. Their random incorporation into EVs appears plausible but active sorting mechanisms have been put forward [17,39–45], while a very recent publication proposed an interesting LC3 driven process [46]. The presence of nuclease-resistant extracellular miRNAs in body fluids

and in supernatants of many different cell types cultivated *in vitro* led to the conclusion that they are contained within the lumen of EVs where they are protected from degradation by ubiquitous RNase activities.

In contrast to this hypothesis, two independent groups found that the majority of circulating miRNAs is mostly complexed with RNA binding proteins such as AGO2 that also prevent degradation by RNases [10,20]. The molar concentration of miRNAs found in EVs in our work and the work of others are far below assumed functional levels, which was estimated to be on the order of 100 copies per cell, depending on the abundance of mRNAs target molecules [47–49], even when all EVs fused with cell membranes of recipient cells delivering their miRNA cargo [21,50,51]. Therefore, it appeared to be unlikely that EV-contained miRNAs can modulate target transcript levels in recipient cells.

We revisited this obvious conflicting issue at three functional levels. First, we made use of EBV's biology and engineered LCLs (i.e., human B-cell lines that differ only in their capacity to encode viral miRNAs). Pairs of EV preparations from the same cell type with and without viral miRNAs allow to distinguish between direct effects of miRNAs versus high doses of EVs. We purified EVs released from EBV-infected cells with two different methods, based on their physical density (Figs 1 and 2) and size (Fig 3). We analyzed their physical concentration by NTA and quantified the contained miRNAs. In our biochemical approach, less than 5% of all miRNAs co-purified with EVs (Fig 3), and the majority of miRNAs remained in the supernatant and did not pellet together with EVs (Fig 2), supporting previous findings [10,20].

Second, we employed a dual luciferase reporter system, consisting of three complementary sequences of the miRNAs of interest downstream of the Renilla luciferase reporter to allow a sensitive detection of the activity of miRNAs (Figs 6 and S6A). Using this approach, we failed to observe viral miRNA activity in recipient cells incubated with EVs, suggesting a failure to transfer functional numbers of miRNAs via EVs in this assay.

Third, we developed a sensitive assay that detects fusion events between EVs and their presumed acceptor cells at a single-cell level (Fig 5). EVs did fuse with various primary cells contained in PBMCs or established cell lines representing different cell types but only when equipped with a fusogenic protein, such as VSV-G (Fig 5). The intrinsic capacity of EV to fuse with either plasma membranes of recipient cells or endosomes after receptor-mediated uptake was extremely low (Fig 5F) or undetectable (Fig 5C and 5E) even when using high EV per cell ratios. Only when EVs were assembled with VSV-G, as few as 1000 EVs sufficed to generate a robust specific fusion signal in a single cell (Fig 5D). Using VSV-G to foster fusion of EVs with cells, we also observed a remarkable delivery of mRNA transcripts coding for two different luciferase reporters (Fig 6B). However, this approach failed to demonstrate an EV-mediated transfer of functional single miRNA species (Fig 6C).

EVs from EBV-infected cells physically interact with many cell types (Fig 4)[14,15,22]. To reevaluate these reports, we stained EVs purified from EBV-infected LCLs with PKH26 dyes. Despite using high numbers of EVs (>1000/cell), cells of different lineages did not stain uniformly in these experiments, suggesting a specific interaction between EVs and certain cell types (Fig 4C). In fact, even though apparently robust interaction between EVs and various target cells could be observed, an EV-mediated transfer of material was extremely inefficient or was undetectable, according to our newly developed fusion assay (Fig 5C). Many papers report the successful EV-mediated transfer of miRNAs to different cells regulating intercellular communication as summarized in a recent review [52]. Our findings strongly argue against a functional transfer of cargo from cell to cell by EVs.

Different methods of preparing EVs might explain this apparent discrepancy. Studying EVs released from different cells is demanding. EVs are small, abundant and probably heterogeneous, and their study requires accurate purification methods and a repertoire of sophisticated

instruments, such as nanoparticle tracking analytics, flow cytometry devices among others, as well as standard procedures [53]. Still, EVs are difficult to enumerate, and biochemical preparations of EV often contain exogenous vesicles as most cells have to be cultivated with serum components of bovine origin. We took great care to deplete EVs and eliminate protein aggregates from our cell-culture components (S1 Fig).

An even more demanding challenge is the analysis of EV-mediated transfer of cargo. More recent approaches include the labeling of EVs with fluorescent dyes or proteins to document protein internalization and luciferase activity in recipient cells *in vitro* ([54] for a recent review). A versatile genetic approach makes use of CRE mRNA in the engineered EVs, which is translated upon delivery so that functional CRE protein throws a genetic switch in acceptor cells [55,56]. However, free CRE protein is taken up by cells [57] even when it lacks a transduction domain [58], potentially causing false positive results. A most recent development uses a CRISPR-Cas9-based reporter to document gene editing in recipient cells upon EV-mediated sgRNA transfer [59]. By these methods, the transfer of EV content is extremely inefficient [55,56,59]. Yet another very recent approach used a split luciferase NanoBiT system and demonstrated that a delivery of EV cargo is only detectable, when EVs are assembled with VSV-G [60], which is in agreement with our findings reported here.

As some of these techniques seem laborious or prone to false positive results, we developed a novel system that monitors the fusion of EVs with membranes of acceptor cells. This novel 'EV fusion assay' is based on an established method used in the HIV field to study viral entry [61]. In this pioneering approach, β-lactamase is fused with the viral protein vpr to trap the enzyme in viral infectious particles. Upon infection viral and cellular membranes fuse and β-lactamase is delivered into the cytoplasmic compartment of HIV-infected cells where it cleaves its substrate [29,61]. We adapted this assay to deliver β-lactamase via EVs fusing the β-lactamase gene with CD63, a surface protein that is predominant in EVs. Stable or transient expression of CD63-BlaM in any cell is feasible, non-toxic and readily gives rise to CD63-BlaM-assembled EVs. ß-lactamase as the enzymatic moiety in this setting is advantageous, because only donor cells have to be engineered to deliver CD63-BlaM whereas any class of target cells including primary cells of interest can be tested (Fig 5E and 5F). In contrast, other methods including the split luciferase NanoBiT system requires the manipulation of both donor and recipient cells [60]. The EV fusion assay can be analyzed by flow cytometry or fluorescent microscopy, two easy and fast read-outs. The assay is reliable, quantifiable and free of background noise in combination with flow cytometry (Fig 5B).

Applying this assay (described in detail in Fig 5A), we learned that EVs equipped with a membrane-targeted (i.e., CD63-fused) ß-lactamase protein (Fig 5E) are taken up by monocytes, B lymphocytes and pDCs derived from PBMCs but only at a very low rate (Fig 5F). Only when EVs were engineered to contain a virus-derived fusogenic glycoprotein, we found a very efficient transfer of ß-lactamase activity to target cells, indicating that as few as $10^3$–$10^4$ EVs per cell sufficed to detect EV-to-cell fusion events 4 hours post-incubation (Fig 5D). An incubation period longer than 4 hours did not lead to a marked signal increase (S4B Fig). The EV field is still developing and standardized purification and analytical methods do not yet exist [53], but the ß-lactamase fusion assay has the potential to become a routine method for quantitating the uptake of EVs by acceptor cells in an unequivocal, background-free and quantitative manner. Recently, we successfully adapted this assay to characterize the fusion of EBV particles with different primary human immune cells [62] underlining the flexibility of our method.

The NTA approach allows the physical quantitation of EVs [50,63], which led to the conclusion that the majority of EVs are free of miRNAs [21]. Our findings recapitulate this notion in EV preparations from the supernatants of LCLs (Figs 3 and 7), arguing that the scarcity of miRNA containing EVs strongly speak against their role as vectors for cell-to-cell

communication [10,64]. Moreover, all our attempts to detect a functional miRNA transfer to recipient cells equipped with a highly sensitive detector failed (Figs 6 and S6C). Even when EVs with an engineered fusogenic moiety and a high rate of EV-to-cell transduction were employed, we could not find a convincing functionality of EV-borne miRNAs (Fig 6C) very much in contrast to EVs engineered to carry functional mRNAs encoding luciferase genes in a reciprocal setting (Fig 6B).

We designed our experiments to investigate and quantitate the functions of EBV-encoded miRNAs in EVs, but the experiments resulted in a series of negative results as presented in this work. This outcome compares to our published work with virus particles released from EBV-infected cells that support the productive, lytic phase of this virus [35]. EBV particles do contain viral miRNAs at higher numbers than EVs released from latently EBV-infected cells and, very much in contrast to EVs, EBV particles are highly infectious and efficiently target primary immune cells, such as B cells among other cells. Preliminary experiments suggest that viral miRNAs contained in EBV particles can have a direct function in targeted cells where they regulate innate immunity [62].

Certain caveats remain. For example, in our study we could use a finite number of cell lines, only, which release EVs, but in many experiments, we made use of newly established lymphoblastoid cell lines (i.e., primary B cells immortalized by EBV) to produce and characterize miRNAs released by these cells. This is a well-accepted model [14,27], which–as far one knows–resembles as much as possible EVs released by EBV infected B cells *in vivo*. The viral miRNAs are expressed from their natural genetic loci in the EBV genome suggesting that their expression is under the control of viral promoters, which are also active *in vivo*. Moreover, these miRNAs have been already reported to be enriched and functional in EVs released by LCLs [14,15]. These earlier reports were our initial motivation to look into this phenomenon and characterize the functions of the viral miRNAs further.

Clearly, all our experiments were done *in vitro* using artificial settings. We did not include senescent cells as recipients or primary human postmitotic cells such as e.g. neurons, but also resting PBMCs did not take up EVs as shown in Fig 5F. Another concern are virus miRNAs, which might not entirely reflect the host cell-secreted miRNAs contained in EVs. A single human miRNA, hsa-miR-16, served as cellular reference in our study, but its functions were indistinguishable from viral miRNAs. We also did not consider that some host-derived miRNAs might be tailed or processed to facilitate their sorting and enrichment into extracellular vesicles [65], but these miRNAs are exceptions regarding their biogenesis and further processing and sorting.

In conclusion, our findings demonstrate that the delivery of different species of RNAs as well as proteins through the EVs is an extremely inefficient process. Although we examined a limited number of miRNAs, only, our data strongly argue against a biologically active transfer of miRNAs as well as of EV cargo in general and at physiologically relevant level, as claimed by others. Whether and to what extent the EVs released from different donor cell types, other than the ones tested here, do carry and specifically release their cargo in a paracrine manner to diverse types of recipient cells needs to be investigated. In this study, we provide and propose several new types of positive and negative controls, as well as different assays that can be easily employed to address these questions in virtually any possible model.

## Materials and methods

### Cell lines and cell culture

EBV-infected primary human B cells, the resulting lymphoblastoid cell lines (LCLs), peripheral blood mononuclear cell (PBMCs), the monocytic cell lines THP-1, the EBV-positive Burkitt

lymphoma cell line Raji and the HEK293-based EBV producer cell lines were maintained in RPMI medium 1640 (Life Technologies). HEK293T cells were maintained in DMEM (Life Technologies). All media were supplemented with 10% FCS, penicillin (100 U/mL; Life Technologies), and streptomycin (100 mg/mL; Life Technologies). Cells were cultivated at 37˚C in a water-saturated atmosphere with 5% $CO_2$. Cell viability was checked by trypan blue exclusion and cultures with more than 95% viable cells were used for experiments.

## Separation of human primary lymphocytes

Human primary B cells were prepared from adenoidal mononuclear cells by Ficoll Hypaque (PAN Biotech) gradient centrifugation as described in [66]. B cells were isolated using CD19 MicroBeads and MACS separation columns (Miltenyi Biotec), according to the manufacturer's instruction.

## Isolation of extracellular vesicles (EVs)

We submitted all relevant data of our experiments to EV-TRACK knowledgebase (EV-TRACK ID: EV200039) (EV-TRACK et al., 2017). EVs were isolated from the supernatant of EBV-infected B cells incubated for 72 hours with medium containing EV-free FCS. We developed the following process to avoid the carry-over of bovine serum-derived EVs to the cell culture medium. FCS was diluted 1:1 with RPMI and centrifuged at 100,000 g at 4˚C in a swinging-bucket rotor (SW28 or 32, Beckman Coulter) for 18 h. The supernatant was filter and sterilized using a 0.22-µm mesh size filter (Sartorius) and then filtered using a 300K Vivaspin 20 (PES, Sartorius) device at 2,000 g at 10˚C for 20–30 min. The EV-free FCS was tested for proteins and particle content (S1 Fig), aliquoted and stored at -80˚C.

According to the protocol shown in Fig 1A, EBV-infected B cells were washed with PBS twice and seeded at a density of $0.5×10^6$ cells/ml. After 72 hours, the conditioned cell-culture medium was centrifuged at 300 g at 20˚C for 10 min, followed by a second centrifugation at 2,000 g at 4˚C for 20 min to remove cells and debris, respectively.

To generate EVs from 293T cells in Fig 6, $1×10^7$ cells were seeded in a 13-cm dish. The next day, the cells were transfected with 12 µg of plasmid DNA encoding one of the two luciferase reporter plasmids or one of the two viral miRNAs (ebv-miR-BART1 or ebv-miR-BHRF1-2) with or without 8 µg of a VSV-G expression plasmid as indicated in panels B and C, respectively. At 24 hours after transfection, the medium was replaced with fresh DMEM without FCS. After 72 hours, the conditioned cell-culture medium was collected and processed in the same way as conditioned medium from EBV-infected B cells.

The supernatant was filtered by using a 0.45-µm filter (Millipore) and EVs were pelleted by ultracentrifugation (UC) in a 30 ml tube (Kisker Biotech, Cat no UZ-PA-38.5–1) at 100,000 g at 4˚C for 2 h in a swinging-bucket rotor (Beckman Coulter; SW28 or SW32).

The supernatant was completely removed and 500 µl of sterile-filtered PBS supplemented with protease inhibitors (Roche, Cat no 11836170001) were added to the bottom of each tube. To resuspend the EVs in the 'UC pellet', the tubes were incubated on ice under agitation for 30 min. The resuspended EVs were then transferred to a 1.5-ml ultracentrifuge tube (Beckman Coulter, Cat no 357448) and centrifuged at 160,000 g at 4˚C for 1.5 h in a fixed-angle rotor to obtain the sediment termed 'miniUC pellet'. The supernatant was completely removed and the miniUC pellet was resuspended in filtered PBS. The final volume depended on the subsequent steps. For standard preparation, we started with 180 ml of cell-culture medium to reach a final volume of 1 ml (180-fold concentration). For further purification and analysis, 380 µl of the miniUC pellet preparation was loaded at the bottom of an iodixanol (Optiprep; Sigma, Cat no D1556) discontinuous density gradient in ultra-clear centrifuge tubes (Beckman Coulter, Cat

no 344062). The density gradient was prepared as follows: 380 μl of the EV sample was mixed with 520 μl of iodixanol (Optiprep) (60%) and placed at the bottom of the tube. Then 2.5 ml of a 1:1 dilution of iodixanol and PBS (30% final concentration of Optiprep) was placed on top to form the middle layer fraction of the gradient. The top layer with a volume of 600 μl consisted of filtered PBS to obtain a final total volume of 4 ml. The density gradient was centrifuged at 100,000 g at 4˚C for 18 h in a SW60Ti swing-out rotor. In general, 10 fractions (400 μl each) were collected starting from the top. The refractive density of the fractions was measured with a refractometer (Abbe Mark III, Reichert Technologies).

## Labelling of EVs with PKH26 membrane dye

EVs were isolated by ultracentrifugation as described above. The EV pellet was resuspended in 200 μl of filtered PBS and stained by using the PKH26 Red Fluorescent Cell Linker Kit for General Cell Membrane Labelling (Sigma-Aldrich). The dye solution was freshly prepared by adding 4 μl of PKH26 dye to 1 ml of Diluent C. The EV preparation was mixed with 1 ml of Diluent C before the dye solution was added. The mixture was incubated at room temperature for 5 min with periodic mixing. Then an equal volume of sterile-filtered 1% BSA was added and incubated for 1 min to stop the staining reaction. To wash the sample and reduce its volume, the stained EVs were mixed with PBS and loaded on a 15-ml Amicon Ultra-15 centrifugal filter of 100K cutoff (PES; Millipore) and centrifuged at 2,000 g for 20–30 min at 10˚C until the volume was reduced to about 400 μl. Then 380 μl of stained sample were loaded at a bottom of an iodixanol (Optiprep) discontinuous density gradient. Fractions containing the stained EVs (2 and 3) were collected and washed three times with 10 ml of filtered PBS in a 100K Amicon Ultra-15 centrifugal filter at 2000 g for 10–20 min at 10˚C. Concentrated stained EVs were resuspended in 1 ml of RPMI containing EV-free FCS.

## Nanoparticle tracking analysis

Nanoparticle tracking analysis (NTA) was performed with the ZetaView PMX110 instrument (ParticleMetrix) and the corresponding software (ZetaView 8.04.02) was used to measure the number and the size distribution of the vesicle preparations. Samples were diluted in filtered PBS to achieve a particle concentration in between $1 \times 10^7$ to $1 \times 10^8$ particles/ml. Within this range, we confirmed the linearity of data acquisition with beads of known concentration and size (102.7±1.3 nm, Polysciences) comparable with EVs (S2 Fig). We also determined the range of accuracy of absolute particle quantification by NTA to be in the range of $1 \times 10^7$ to $1 \times 10^8$ particles/ml, again using dilutions of calibration beads of known initial particle concentration (S2 Fig). All samples were measured at 11 positions with three reading cycles at each position. Pre-acquisition parameters were set to a sensitivity of 75, a shutter speed of 50, a frame rate of 30 frames per second and a trace length of 15. The post-acquisition parameters were set to a minimum brightness of 20, a minimum size of 5 pixels and a maximum size of 1000 pixels.

## Size exclusion chromatography (SEC)

SEC was performed using qEV columns (Izon science), following the manufacturer's protocol. EBV-infected B cells were washed twice with PBS and seeded at a density of $0.5 \times 10^6$ cells/ml in RPMI, supplemented with 2% EV-free FCS, penicillin (100 U/mL), and streptomycin (100 mg/mL). After 72 h, the conditioned cell-culture medium was centrifuged at 300 g for 10 min at 20˚C, followed by a second centrifugation at 2,000 g for 20 min at 4˚C. The supernatant was then concentrated 10-fold using the 100K Amicon Ultra-15 centrifugal filter at 2,000 g for 10–20 min at 10˚C. 1 ml of the concentrated supernatant was used for SEC and 20 fractions

(500 μl for each) were collected. 800 μl of Qiazol (Qiagen) was added to 300 μl of each fraction for RNA extraction. 10 μl of each fraction were used for western blots (WBs). All parameters of preparing EVs are available at EV-TRACK knowledgebase (http://evtrack.org/review.php) with the EV-TRACK ID EV200039.

## Protein extraction and Western blot analysis

Cells were lysed with RIPA buffer [50 mM Tris·HCl (pH 8), 150 mM NaCl, 0.1% SDS, 1% Nonidet P-40, 0.5% DOC], kept 30 min on ice and then stored at -80˚C. Protein concentrations were quantified with the Pierce BCA protein assay kit (Thermo Scientific), following the manufacturer's protocol. 5–20 μl of samples were loaded per well, depending on the source of the proteins. The absorbance was measured at 550 nm in an EL800 Universal microplate reader (BioTek instruments). For samples from SEC, RIPA buffer was not used and the protein concentration was measured with Bradford assay (Millipore).

Absorbance values of the unknown samples were determined by the interpolation with the BSA standard values using a four-parameter logistic (4PL) curve (Graphpad). Cell lysates or proteins from the different EV preparations were diluted in Laemmli buffer and denatured at 95˚C for 5 min. Proteins were separated on 10–12% SDS- polyacrylamide gel electrophoresis gel (SDS-PAGE; Carl Roth) and transferred to nitrocellulose membranes (GE Healthcare Life Science). Membranes were blocked for 1 h with 5% non-fat milk in PBS-T (PBS with 0.1% Tween-20) and incubated with the antibody of interest. Secondary antibodies conjugated with horseradish peroxidase were used and exposed to CEA films (Agfa HealthCare). SDS-PAGE was stained with Coomassie (Serva Blue G-250) for 5 min and destained with a solution of 15% acetic acid in $H_2O$. The following antibodies were used for WB: mouse anti-TSG101 1:2000 (4A10; Genetex), mouse anti-Calnexin 1:1000 (610523; BD Bioscience), and rat anti-LMP1 (1G6-3; Helmholtz Zentrum München).

## RNA extraction

RNA was isolated using the miRNeasy Kit (Qiagen), according to the manufacturer's protocol. Pellets of cells or EVs were resuspended in 700 μl of Qiazol (Qiagen), whereas 800 μl of Qiazol were added to 200–300 μl of liquid samples, such as cell supernatants or SEC fractions. Samples were vortexed for 10 sec, incubated at RT for 5 min, frozen on dry ice, and stored at -80˚C.

For RNA extraction, samples were thawed on ice. 2 μl of RNA-grade glycogen (Thermo scientific) were added to improve RNA recovery. We also added 10 μl ($10^7$ copies) of the synthetic *Caenorhabditis elegans* miRNA cel-miR-39 (5′-UCACCGGGUGUAAAUCAGCUUG-3'; Metabion) as spike-in control RNA since it has no mammalian homologue. 0.2 volumes of chloroform were added to each sample and mixed by vortexing for 10 sec. We then followed the manufacturer's protocol, including also the optional step of washing with the RWT buffer. The RNA was resuspended in 30 μl of nuclease-free water (Peqlab) in Eppendorf LoBind microcentrifuge tubes.

## RNA quantification and quality analysis

RNA was quantified with a NanoDrop 1000 spectrophotometer (Thermo Scientific), the quality of the RNA from cells and EVs was assessed with an Agilent 2100 Bioanalyzer (Agilent), according to the manufacturer's protocol (RNA 6000 Pico Kit).

## Stem-loop quantitative RT-PCR of microRNAs

Single-stranded cDNA synthesis was performed for each single miRNA of interest using the TaqMan MicroRNA Reverse Transcription kit (Thermo Scientific), following the manufacturer's protocol. For samples with high amount of RNA, such as cell lysate, 200 ng of RNA was used. For samples with low yield of RNA, such as EVs preparations, RNA was diluted 1:3 with nuclease-free water, and 9 µl of this dilution was used for the reverse-transcription (RT). After the RT, the samples were diluted five times with nuclease-free water and 4 µl of these dilutions were applied for qPCR using the TaqMan Fast Advanced Master Mix (Thermo Scientific).

The following TaqMan MicroRNA assay were used: ebv-miR-BHRF1-2-3p (197239_mat); ebv-miR-BHRF1-1 (007757); ebv-miR-BART1-5p (197199_mat); ebv-miR-BART3 (004578_mat); hsa-miR-16 (000391); and cel-miR-39 (000200).

For the absolute quantification of mature miRNAs, RNA oligomers corresponding to the mature miRNA sequences of interest were synthesized (Metabion). The quality and quantity of the synthetic RNA molecules were confirmed using an Agilent 2100 Bioanalyzer and a NanoDrop 1000 spectrophotometer (S3 Fig), respectively. A dilution of the synthetic RNAs was prepared freshly immediately before each miRNA quantification. RNA oligos were diluted with nuclease-free water to concentrations ranging from 10 to $10^8$ copies per ml. Quantitative RT-PCR was performed in parallel and a standard curve of $C_T$ (threshold cycle) values was calculated using a four-parameter logistic (4PL) fit (Graphpad) for the interpolation of sample $C_T$ values. Absolute copy numbers of *C. elegans* cel-miR-39 were used as spike-in miRNA to normalize samples.

## Luciferase reporter assays

Three tandem repeats of a miRNA binding site (3x PT) for the EBV miRNA of interest were cloned into the downstream the *Renilla* luciferase (*Rluc*) in a psiCHECK2 plasmid (Promega). The following primers were used:

miR-BART1-5p Forward

5'-**TCGA**GCACAGCACGTCACTTCCACTAAGAAATTCACAGCACGTCACTTCCACT
AAGAAATTCACAGCACGTCACTTCCACTAAGAGC-3'

miR-BART1-5p Reverse

5'-**GGCC**GCTCTTAGTGGAAGTGACGTGCTGTGAATTTCTTAGTGGAAGTGACGTG
CTGTGAATTTCTTAGTGGAAGTGACGTGCTGTGC-3'

miR-BART3-3p Forward

5'-**TCGA**GACACCTGGTGACTAGTGGTGCGAATTACACCTGGTGACTAGTGGTGCG
AATTACACCTGGTGACTAGTGGTGCGGC-3'

miR-BART3-3p Reverse

5'-**GGCC**GCCGCACCACTAGTCACCAGGTGTAATTCGCACCACTAGTCACCAGGTG
TAATTCGCACCACTAGTCACCAGGTGTC-3'

miR-BHRF1-2 Forward

5'-**TCGA**GTCAATTTCTGCCGCAAAAGATAAATTTCAATTTCTGCCGCAAAAGATAA
ATTTCAATTTCTGCCGCAAAAGATAGC-3'

miR-BHRF1-2 Reverse

5'-**GGCC**GCTATCTTTTGCGGCAGAAATTGAAATTTATCTTTTGCGGCAGAAATTGA
AATTTATCTTTTGCGGCAGAAATTGAC-3'

The primers were annealed and ligated with psiCHECK2 digested with XhoI and NotI (NEB). The pCDH vectors harboring single EBV miRNAs were used as described [67]. 293T cells were seeded in a 24-well plate at a density of $1 \times 10^5$ cells/well. After 24 h, cells were co-transfected with 30 ng of specific miRNA reporter plasmid (3x PT psiCHECK2) and different amounts of pCDH vector expressing a miRNA of interest (0.39–100 ng). A pCDH empty vector was added to compensate the different amounts of vector used to reach 100 ng in each condition.

Alternatively, extracellular vesicles were used as source of miRNAs. 4 or 24 h after transfection different amounts of EVs isolated from WT or ΔmiRNA EBV-infected B cells were added. 100 μl corresponds to $1 \times 10^{10}$ EVs, if not indicated differently. At 24 h after DNA transfection, luciferase activities were determined with the Dual-Luciferase Assay Kit (Promega) and the Orion II Microplate Luminometer (Titertek-Berthold). The activity of Rluc was normalized to the activity of *Firefly* luciferase (*Fluc*) encoded by the psiCHECK2 plasmid.

## MTT assay

EBV-infected B cells were seeded at an optimal concentration of $0.5 \times 10^6$ cells/ml with different media conditions. After 72 h, viable cells were assessed by an MTT assay [68].

## CellTiter-Glo

293T cells or EBV-infected B cells were seeded in a 24-well plate at a density of $1 \times 10^5$ cells/well. After 24 h, different amounts of EVs isolated from WT EBV-infected B cells were added. At 8 or 24 h after incubation, cell viability was assessed by CellTiter-Glo 2.0 (Promega), following the manufacturer's protocol. Luminescence signals were measured by the Orion II Microplate Luminometer (Titertek-Berthold).

## FACS analysis of PBMCs

Isolated PBMCs were stained with the following antibodies:

CD19 –APC (Clone: HIB19; 17-0199-42, BD Biosciences)

CD3 –APC (Clone: SP34-2; 557597, BD Biosciences)

CD11c –APC (Clone: 3.9; 301613, BioLegend)

CD14 –PE (Clone: MEM-15; 21279144, ImmunoTools)

CD303 –APC (Clone: AC144; 130-097-931, MACS)

CD304 –PE (Clone: REA774; 130-112-045, MACS)

CD123 –VioBlue (Clone: AC145; 130-113-891, MACS)

## ß-Lactamase (BlaM)-based fusion assay

The expression plasmid p7200 encoding CD63-BlaM was constructed based on pcDNA3.1(+). It encodes the open reading frame of human CD63, which is carboxy-terminally fused (via a $G_4S$ flexible linker) to a codon-optimized ß-lactamase (BlaM) gene. The CD63-BlaM expression cassette was moved into the context of a lentiviral vector backbone, based on a pCDH derived vector (System Biosciences) so that mtagBFP, an enhanced monomeric blue fluorescent protein, is co-expressed together with CD63-BlaM. The lentiviral vector was termed p7250.

To generate CD63-BlaM assembled EVs from 293T cells $1\times10^7$ cells were seeded in a 13-cm dish. After overnight incubation, 12 μg of CD63-BlaM plasmid DNA p7200 was chemically complexed and transfected alone or together with 8 μg of the VSV-G expression plasmid p5451. The next day, the medium was exchanged with non-supplemented, plain DMEM cell culture medium with 5 g/L D-glucose. After 72 h, the supernatant (conditioned medium) was harvested. EVs were isolated by serial centrifugation and density gradient as described above.

To generate CD63-BlaM assembled EVs from stably transduced LCLs and 293T cells, the cells were transduced with the lentiviral vector p7250, which expresses mtagBFP, an enhanced monomeric blue fluorescent protein, a T2A element, and the open reading frame encompassing CD63-BlaM. Lentivirally transduced cells were enriched by FACS according to their highest expression of mtagBFP. The cells were seeded at an initial density of $5\times10^5$/ml and the conditioned medium was harvested 72 h later. EVs were isolated by serial centrifugation and density gradient as described above.

The conditioned cell-culture medium or purified EVs from donor cells were incubated with $2\times10^5$ recipient cells at 37°C for 4 h. Cells were washed, trypsinized and collected at 500 g for 10 min. Cells were resuspended with 100 μl of CCF4-AM staining solution in a 96-well plate. The staining solution per ml consisted of 1 ml of $CO_2$-independent cell-culture medium (Thermo Fisher Scientific, Cat no 18045), 2 μl of CCF4-AM, 8 μl of Solution B (Thermo Fisher Scientific, Cat no K1095) and 10 μl of 250 mM Probenecid (Sigma, Cat no P8761). Cells were incubated at room temperature in the dark for 16 h. Subsequently, the cells were washed with PBS twice. 293T cells were fixed with 4% PFA (Merck, Cat no 104005) for 30 min, whereas PBMCs were processed further omitting the fixation step. The measurement was performed by flow cytometry using an LSR Fortessa instrument (BD). The 409-nm wavelength laser (violet) was used for excitation of the FRET substrate, and the emission of the intact, non-cleaved CCF4 substrate was detected at 520 nm (green), whereas the emission of the cleaved CCF4 substrate was detected at 447 nm (blue).

### EBV production and virus titration with Raji cells

The recombinant EBV genomes used in this study are the plasmid p2089 [69], a wild-type (WT) EBV, and the p4027, knockout for all the viral miRNAs (ΔmiR EBV) [23].

Virus used to infect primary B cells was produced as described [23]. Briefly, HEK293-based producer cell lines, which stably carry recombinant EBV genomes, were transfected with plasmids coding for the viral proteins BZLF1 and BALF4 to induce the viral lytic phase. Supernatant was collected 3 days after transfection and titrated using Raji cells as described [23,70,71]. Isolated primary B cells were infected with a multiplicity of infection of 0.1 Green Raji Units (GRU). At 18 h later, the infected B cells were washed and cultivated at an optimal initial density of $5\times10^5$ cells/ml.

### Confocal microscopy

The adherent cell lines U-251MG, MDA-MB-231 and LN-18 were seeded on glass coverslips (Carl Roth) coated with fibronectin (Advanced Biomatrix). After 24 hours, medium was replaced and CD63-BlaM assembled EVs with VSV-G purified from 293T were added to the cells. Cells untreated are used in parallel as negative control. Four hours later cells were washed three times with PBS, loaded with CCF4-AM and developed overnight as described above. Subsequently, cells were washed three times with PBS, fixed with 4% PFA for 10 min at room temperature and washed again. The coverslips were mounted with ProLong Diamond Anti-fade Mountant (Thermo Fischer Scientific) and were analyzed with a spinning disk confocal microscope (Nikon) using a 60x objective lens. The Imaris software was used to analyze the

images. This filter configuration used led to a slight spillover of the cleaved CCF-4 (blue channel) into the CCF-4 (green) channel (see S5B Fig).

## Negative stain transmission electron microscopy (TEM)

EVs released from LCLs were collected after iodixanol density gradient centrifugation (fractions 2 and 3) and 100K Amicon Ultra-15 centrifugal filters were used to transfer the EVs to HEPES-Buffered Saline (HBS). Aliquots of 5 μl of EV preparations in HEPES buffered saline at an approximate concentration of $3.4 \times 10^{13}$ particles per ml were placed on glow-discharged continuous carbon film supported copper grids (3 mm, 300 mesh, Plano) and adsorbed for 5 min. After sample removal, grids were stained for 30 sec with uranyl acetate (2% w/v). Micrographs were imaged with a 40,000 fold magnification (0.414 nm/pix) using a JEOL JEM-1400 Plus microscope operating at 120 kV with a JEOL CCD Ruby camera. The underfocus was set to 500 nm.

## Supporting information

**S1 Fig. Preparation of fetal calf serum (FCS) to reduce the concentration of bovine EVs.**
FCS was diluted 1:1 with RPMI1640 medium and centrifuged at 100,000 g at 4˚C in a swinging-bucket rotor (SW28 or 32, Beckman Coulter) for 18 h. After ultracentrifugation (UC), the supernatant was further processed using ultrafiltration spin columns purchased from different manufacturers and with different cutoffs: 100K Amicon Ultra-15 (ultracel regenerated cellulose, Merck), 300K Vivaspin 20 or 1000K Vivaspin 20 concentrators (PES, polyethersulfon, Sartorius). The columns were centrifuged at 2,000 g, 10˚C for 20–30 min. The different preparations were tested for their EV concentrations by NTA, protein content and adverse effects on a lymphoblastoid cell line (LCL). **(A)** EV particle concentrations of RPMI supplemented with 10% untreated, not EV-depleted fetal calf serum (FCS) or supplemented with 10% FCS treated with three different centrifugal filter were measured by NTA with the ZetaView instrument PMX110 to analyze the EV concentrations after UC and followed by ultrafiltration as indicated. A single UC step reduced the concentration of EVs to about 20%, whereas ultrafiltration with 100K or 300K filters removed the majority of bovine EVs. **(B)** Protein concentrations (as measured by Bradford) of the samples analyzed in panel A. Filtration with the 100K filter device led to a considerable reduction of proteins contained in FCS. **(C)** LCLs were cultivated in RPMI1640 supplemented with 10% FCS conditioned as shown in panel A. After 3 days, an MTT assay was performed to assess the viability of the cells. As a negative control, LCLs were treated with 10 μM etoposide for 1 h to induce cell death. Cells cultivated in RPMI1640 with 10% FCS passed through the 300K centrifugal filter device after UC scored better in this assay than cells cultivated with 10% FCS using the 100K centrifugal filter. **(D)** LCLs were cultivated as in panel C but with different % of FCS filtered with 300K centrifugal filters. Results with B cells from two independent donors are shown. RPMI1640 medium supplemented with 2% FCS after UC and 300K ultrafiltration was used in all experiments throughout the manuscript.
(PDF)

**S2 Fig. Validation of the ZetaView PMX110 instrument used for nanoparticle tracking analysis (NTA).** To assess the accuracy and sensitivity of physical particle measurement with the ZetaView instrument (ParticleMetrix), we used calibration beads of known size (102.7±1.3 nm, Polysciences, Cat. #64010) in the range of EVs. **(A)** Serial dilutions of beads ranging from $1 \times 10^5$ to $1 \times 10^7$ per ml were performed, and each dilution was measured in the ZetaView instrument to confirm the linearity of data acquisition and measurement within this range.

**(B)** Accuracy of absolute particle quantification was determined by NTA using dilutions of the calibration beads as in panel A of known initial particle concentration. Mean and standard deviation of three independent replicates are shown.
(PDF)

**S3 Fig. Absolute quantification of miRNAs.** For absolute quantification of mature miRNAs, synthetic RNA oligomers ('mimics') corresponding to the mature miRNA sequences of interest were purchased (Metabion). **(A)** The quality of the synthetic RNA molecules was confirmed using an Agilent 2100 Bioanalyzer. Shown is an example of the viral miRNA ebv-miR-BHRF1-2. **(B, C)** Two alternative methods to measure the absolute concentrations of the synthetic miRNAs were compared. **(B)** Single synthetic miRNAs oligomers were diluted first with nuclease-free water to concentrations ranging from $10^0$ to $10^8$ copies per vial followed by reverse transcription (RT) reactions for each vial as described in Materials and Methods. **(C)** Reverse transcription of miRNAs RNA oligomers was performed first, and then the samples were diluted with nuclease-free water to concentrations ranging from $10^0$ to $10^8$ copies. **(D)** After reverse transcription of the miRNAs RNA oligomers as described in panels B or C, quantitative PCRs of each dilution step were performed with three miRNAs of interest: ebv-miR-BART1-5p, ebv-miR-BHRF1-2-3p, and ebv-miR-BART3. The two different approaches of absolute quantification of synthetic miRNAs delivered almost identical results. To mimic the experimental situation, we decided to use the approach shown in panel B in which miRNA samples are diluted prior to reverse transcription. The approach shown in panel C (dilution of a single miRNA sample after reverse transcription) is common, but it does not reflect the situation in samples with only a few miRNA copies in the probes. Mean and standard deviation of three or more independent experiments are shown.
(PDF)

**S4 Fig. EVs do not deliver their cargo to recipient cells unless they carry a fusogenic glycoprotein.** **(A)** 293T, Calu-3, Caco-2, HepG2 and Huh7 were engineered to express CD63-BlaM stably after lentiviral transduction. To boost expression of the CD63-BlaM fusion protein further the cells were transiently transfected with a plasmid expressing the CD63-BlaM fusion protein reporter alone or together with a VSV-G encoding expression plasmid. VSV-G assembled EVs served as positive control. EVs from these cells were purified and incubated with $2 \times 10^5$ cells from 17 different recipient cell lines for 4 h. The cells were loaded with CCF4-AM substrate, fixed and analyzed by flow cytometry. Means of three technical replicates are shown. The summary of the data is shown in a heat map in Fig 5E. **(B)** EVs from the 5 different donor cells were generated as in panel A to carry CD63-BlaM and VSV-G and recipient 293T cells were incubated for 4 hours or for one day (1d). Then, the cells were loaded with CCF4-AM substrate, fixed and analyzed by flow cytometry. The mean of three replicates is shown. Differences between cells incubated with EVs for 4 or 24 hours were minor. **(C)** EVs were purified from five different donor cells and physical concentrations of EVs were determined by NTA. 50 μl of each of these preparations was used in Fig 5E.
(PDF)

**S5 Fig. Details and additional data covering the EV fusion assay.** **(A)** EBV-positive LCL or 293T cells were engineered to express CD63-BlaM stably after lentiviral transduction. VSV-G was transiently transfected into CD63-BlaM-positive 293T cells as indicated in one case. 500 μl of conditioned medium (CM; see Fig 2A) or 50 μl resuspended EVs from the UC pellet (UC) were incubated with $2 \times 10^5$ 293T cells or LCLs as recipients (about 35,000 EVs per cell). The negative control (Ctrl) is a sample of 50 μl EVs obtained from an UC pellet with supernatant of non-transduced, 293T cells. **(B)** The adherent cell lines U-251MG and MDA-MB-231 were

seeded onto glass coverslips (Carl Roth) coated with fibronectin (Advanced Biomatrix). The cells were treated as in Fig 5G but without incubating them with EVs. After 24 hours, medium was replaced with fresh medium and 4 hours later cells were washed three times with PBS and stained with CCF4 overnight. Thereafter, cells were washed three times with PBS, fixed for 10 min with 4% PFA at room temperature and washed again. The coverslips were mounted with ProLong Diamond Antifade Mountant (Thermo Fischer Scientific). The cells were used in parallel as negative controls accompanying Fig 5G. Scale bars is 30 μm.
(PDF)

**S6 Fig. EVs with EBV miRNAs are not functional in 293T target cells. (A)** The design of the modified dual luciferase reporter plasmid based on psiCHECK2 is shown. It encompasses the internal control firefly luciferase (used for normalization) and the reporter Renilla luciferase with three tandem copies of perfect complementary target sites (3xPT) of the miRNAs of interest inserted in the 3'UTR of the Renilla mRNA. **(B)** 293T cells were transfected with 30 ng of the miRNA reporter plasmid containing 3xPT with increasing amounts of the corresponding miRNA expression vector (pCDH) starting with 390 pg up to 100 ng. At 24 h after transfection, cells were lysed to determine the Renilla and firefly luciferase activities. Mean and SD of three replicates are shown. **(C)** 293T cells were transiently transfected with 30 ng of the 3xPT miRNA reporter plasmid. After 4 h, the cells were incubated with increasing amounts of EVs isolated from the supernatants of LCLs infected with wild-type EBV encoding 44 viral miRNAs (WT EV) or infected with ΔmiRNAs EBV, devoid of all viral miRNAs (ΔmiRNA EV). EVs were prepared from the 'miniUC pellet´ (Fig 1A) and resuspended in 100 μl, corresponding to approximately $1 \times 10^{11}$ physical particles per ml as measured by NTA (from 2,500 to 20,000 EVs per cell). After 24 h incubation, the cells were lysed, and Renilla and firefly luciferase activities were measured. One example of three independent experiments is shown.
(PDF)

**S7 Fig. High doses of LCL-derived EVs are toxic to recipient cells. (A)** 293T cells were seeded in a 96-well plate at an initial density of $2.5 \times 10^4$ cells/well. After 24 hours, the cells were transiently transfected with 7.5 ng of a miRNA reporter plasmid (3x PT psiCHECK2). Different amounts of EVs were prepared and concentrated from supernatants of WT (blue) or ΔmiRNA EBV (red)-infected B cells were added, as indicated on the X-axis, 8 h after transfection. 1 μl corresponds to $1 \times 10^9$ EVs as determined by NTA. After a 24-h incubation, the cells were lysed, and Renilla and firefly luciferase activities were measured. **(B)** 293T cells were seeded in a 24-well plate at an initial density of $2.5 \times 10^5$ cells/well. After 24 hours, the cells were treated with different amounts of EVs isolated from supernatants of WT or ΔmiRNA EBV infected B cells as in panel A. 200 μl contain $2 \times 10^{10}$ EVs as measured by NTA. After 24 h, cell viability was measured in an MTT assay (Materials and Methods). Cells not treated with EV (0 μl) were set to 100% for data normalization. Data obtained from one experiment of two independent experiments are shown.
(PDF)

**S8 Fig. About 20–300 miRNAs copies reduce luciferase levels by half.** 293T cells were transfected with 30 ng of the two miRNA reporter plasmids psiCHECK2-ebv-miR-BART1-3xPT or psiCHECK2-ebv-miR-BHRF1-2-3xPT together with increasing amounts of the corresponding miRNA expression vectors (pCDH-miR-BART1 or pCDH-miR-BHRF1-2) as in Fig 6A. After 24 hours, one third of the cells was lysed and used for the luciferase assay, one third was harvested for RNA extraction and miRNAs absolute quantification and one third of the cells was used for absolute cell quantification by flow cytometry using BD Trucount Tubes. Absolute number of miRNAs was divided by the number of cells transfected to estimate the miRNAs/

cell ratio. A single experiment is shown.
(PDF)

## Author Contributions

**Conceptualization:** Manuel Albanese, Takanobu Tagawa, Wolfgang Hammerschmidt.

**Investigation:** Yen-Fu Adam Chen, Anne K. Schütz.

**Methodology:** Manuel Albanese, Yen-Fu Adam Chen, Corinna Hüls, Kathrin Gärtner, Ernesto Mejias-Perez, Christine Göbel, Mikhail Shein, Anne K. Schütz.

**Project administration:** Wolfgang Hammerschmidt.

**Resources:** Manuel Albanese, Oliver T. Keppler, Reinhard Zeidler, Wolfgang Hammerschmidt.

**Supervision:** Wolfgang Hammerschmidt.

**Validation:** Manuel Albanese.

**Visualization:** Manuel Albanese, Yen-Fu Adam Chen.

**Writing – original draft:** Manuel Albanese, Wolfgang Hammerschmidt.

**Writing – review & editing:** Manuel Albanese, Yen-Fu Adam Chen, Kathrin Gärtner, Takanobu Tagawa, Wolfgang Hammerschmidt.

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
