## [Decision Letter · Decision Letter 0]

27 Aug 2021

Dear Dr Hammerschmidt,

Thank you very much for submitting your Research Article entitled 'MicroRNAs are minor constituents of extracellular vesicles that are rarely delivered to target cells' to PLOS Genetics.

We sincerely apologize for the extended delay to solicit reviewers for your manuscript. The manuscript was fully evaluated at the editorial level and by 4 independent peer reviewers, including the three reviewers from Review Commons. The reviewers appreciated the importance of your study, but identified some concerns that we ask you address in a revised manuscript. 

We therefore ask you to modify the manuscript according to the review recommendations, including those already described to be planned in response to the Review Commons reviews. Your revisions should address the specific points made by each reviewer. We understand that you have already addressed several points made by the reviewers from Review Commons, but we note that some were described to be in progress or planned, and should be included in the revised submission. 

[LINK]

Yours sincerely,

Lin He

Associate Editor

PLOS Genetics

John Greally

Section Editor: Epigenetics

PLOS Genetics

Reviewer's Responses to Questions

**Comments to the Authors:**

Reviewer #1: In this manuscript, Albanese M et al. aimed to investigate the capacity of extracellular vesicles (EVs) to deliver packaged miRNAs into target cells and analyze if EV-contained miRNAs could affect the recipient's target mRNA. The authors argue that miRNAs are rarely present in EVs. Unlike several studies, the authors performed a series of experiments, including a new sensitive assay that applied a synthetic, codon-optimized version of the BlaM with CD63 for fusion analysis in the recipient cells, to suggest EV fused with host cells with low efficiency, not to mention to deliver the EV borne miRNAs into recipient cells. Overall, this study provides biochemical evidence to suggest EVs are associated with a low level of extracellular miRNAs. The manuscript is well written and presented. I am intrigued by numerously negative results present in this manuscript to challenge EV-mediated miRNA function to regulate target mRNA in recipient cells. I think the previous three reviewers' comments in the preprint are valid and relevant, and the authors also seem to respond reasonably. Additionally, I suggest a bit more revised experiments and points to address or discuss below:

Major concerns:

1. Most of the recipient cell lines presented in this study are immortal cell lines, which proliferate robustly. This might affect EV fusion and uptake in the recipient cells while under active proliferation. Could the authors try the EBV experiment in more physiological cell settings, such as primary senescent MEFs or postmitotic cells, such as neurons?

2. Luciferase assay is standard and sensitive for testing miRNA targeting, but the stoichiometry between miRNA and targeting also affects the luciferase activity. In Figure1~3, EVs associate with minimal miRNAs, and, predictably, there is no change in Figure 6C. Is that possible to examine the spike in miRNAs in the recipient cells by RNAseq?

3. While I think the biochemical data presented in this study are convincing, the authors did not do a fair job of discussing their data with limitations. For example, the data are presented in limited cell lines, with all in vitro artificial settings. The design of EBV and virus RNA might not entirely reflect the host cell-secreted EVs. Further, some miRNAs might be tailed or with the addition/editing of the nucleotide to facilitate sorting into exosomes, which is not factored in within this study. The authors might need to discuss the shortage of their setting to balance the views of secretory miRNAs.

Minor concerns:

1. The manuscript mentions "data not shown" in the points to the reviewer and the discussion. Please show them.

2. In Figure 1D, how is the RNA percentage quantified? It looks to me that the total RNA is not reaching 100%. Additionally, in Figure 1E, EVs in the negative stain data look very heterogeneous. It will be good to annotate them with arrowheads to indicate which are EVs.

**Have all data underlying the figures and results presented in the manuscript been provided?**

Reviewer #1: Yes

PLOS authors have the option to publish the peer review history of their article (what does this mean?). If published, this will include your full peer review and any attached files.

Reviewer #1: No

---

## [Decision Letter · Decision Letter 1]

1 Nov 2021

Dear Dr Hammerschmidt,

Thank you very much for submitting your Research Article entitled 'MicroRNAs are minor constituents of extracellular vesicles that are rarely delivered to target cells' to PLOS Genetics.

The manuscript was fully evaluated at the editorial level and by independent peer reviewers. The manuscript has improved substantially and most reviewers are satisfied with the new data provided. It is unfortunate that we experienced issues with review commons, such that one reviewer's comments were cut short during the first revision. We did not realize this issue until the previous revision. Hence, we would like to ask you to carefully review the additional comments from reviewer 3. The editors think comments 3, 4, 5, 6 could be an easy fix within a short period, and comment 1 could be completely or partially addressed. We want to assure you that the editors at Plos genetics are quite interested in this story, and we hope you would be willing to spend 2-3 weeks, and address these final comments as best as you can. We sincerely apologize this delay caused by review commons, it is quite unexpected for us at PLOS genetics. 

We therefore ask you to modify the manuscript according to the review recommendations. Your revisions should address the specific points made by each reviewer.

We hope to receive your revised manuscript within the next 21 days. If you anticipate any delay in its return, we would ask you to let us know the expected resubmission date by email to plosgenetics@plos.org.

[LINK]

Yours sincerely,

Lin He

Associate Editor

PLOS Genetics

John Greally

Section Editor: Epigenetics

PLOS Genetics

Reviewer's Responses to Questions

**Comments to the Authors:**

Reviewer #1: The authors have addressed all reviewers' concerns quite nicely. They also provide more evidence to check the EV miRNAs in human PBMC and discuss their limitations for their approach. I would be happy to support their publication.

Reviewer #2: The authors have responded to my concerns.

Reviewer #3: Of note, a major part of my review (section EVIDENCE, REPRODUCIBILITY AND CLARITY) was not transmitted to the author during the reviewing process due to an editor pipeline mistake. In order to inform the authors, and facilitate downstream reviewing process, I thus copy-pasted the missing sections and will then comment on whether these has been fully or partially addressed by the authors during the revision process (e.g. through other reviewer's comments), knowing that the authors were not aware of my initial request

INITIAL REVIEW (MARCH 21)

EVIDENCE, REPRODUCIBILITY AND CLARITY

In this manuscript, Albanese et al are investigating the capacity of extracellular vesicles (EVs) in delivering functional viral-derived miRNA to recipient cells. The key conclusions of the study are that (1) EVs, despite the fact of efficiently associating with the surface of acceptor cells, are not efficiently delivering their luminal cargos in absence of virally-derived fusogenic proteins (VSV-G), (2) in presence of VSV-G, delivery of functional mRNA, but not miRNA, can be observed (3) in this last case, the lack of EV-borne miRNAs functional delivery is likely to be due to their low abundance in EVs fraction. To support their claims, the authors notably repurpose a known virus particle fusion reporter assay for efficiently monitoring EVs fusion, allowing them to test a wide combination of cell lines as sources and targets of EVs. This work will be of high interest to investigators in the field of extracellular vesicles and extracellular RNAs. The study also reports important observation in the field of virology, notably by exploring the potential impact of EVs-borne RNA produced from EBV-infected cells.

The comparison between regular and VSV-G pseudo-typed EVs provides highly convincing evidence of the poor RNA cargo delivery of regular EVs in their experimental system. However, one should note that the miRNA reporter assay (based on luciferase) described in the study is based on transient transfection. Even with a low concentration of plasmids, the level of expression of the luciferase reporter must be high, and it is unclear whether it could be capable of detect low-copy number miRNA functionality (as it would be the case for EV-borne miRNA delivery). This is the main technical caveat of this study.

However, taken globally the results of this study are convincing and are in line with some recent publications demonstrating that EVs fusion and the delivery of their RNA cargo is much less efficient than previously reported. Moreover, the wide array of controls, and the use of the highest standard in the field in terms of EVs isolation (and associated quality controls), provide additional confidence in the validity of their finding.

During the first round of review, I initially suggested two additional experiments to improve the manuscript.

1/ An interesting observation in this manuscript is the demonstration of functional delivery of mRNA, but not miRNA, when using VSV-G functionalized EVs. As commented by the authors, one hypothesis for the lack of EV-borne miRNA is their low abundance in EVs, as reported in many publications in the field. To support this hypothesis, it would be interesting that the authors determine the copy number / EVs of their encapsulated reporter mRNA. Would a higher copy number per EVs than for miRNA explain the increased functionality? Or can it just be explained by the fact that a much lower mRNA copy number is required for their detection assay (as luciferase assays are very sensitive in this context). The authors already produced a batch of VSV-G EVs containing luciferase reporter mRNA, and also demonstrated their capacity to count EVs and absolute miRNA copy numbers from different conditioned medium fractions. These experiments would involve qRT-PCR and RNA spike controls, and as such are relatively inexpensive.

2/ I would suggest the authors demonstrate the sensitivity of their miRNA reporter assay by assembling a reporter for low expressed miRNA (200 copies per cell or less, as initially determined by RNA-seq and absolute miRNA count per cell) and evaluate its performance in this context. Alternatively, establishing a stable line expressing the reporter at a low level, or using lentiviral transduction at low MOI and/or a low activity promoter would allow obtaining more physiologic expression level. Establishing a new reporter for low-abundance miRNA and testing it by transient transfection can be achieved in 1-2 weeks, and at low cost (a few oligos, and cloning steps followed by transient transfection). Establishing stable cell lines expressing viral miRNA reporter at low expression level would be longer (1 month) but could lead to a more sensitive cell-based assay for detecting a low-copy number of EV-borne viral miRNA delivered by EVs.

I also had minor comments that could be addressed by the author to improve the manuscript:

3/ In several figures (at least Fig. 1, 3, 4, S6, S7), the authors present a representative experiment among 2 or 3 executed. It would be better to present graphs representing the average value among replicates (including error bars), or at least to include the additional dataset as supplementary data.

4/ Figure 5E, and S4A: The authors mention that this experiment was replicated 3 times. It is however unclear whether it is technical replicates (EVs prepped only once, but transfer assay conducted 3 times in parallel, or whether this is a biological replicate (multiple EVs preparation, and independent test on acceptor cells). Given the importance and interest of this key experiment, it is important to precise as variability in the assay could impact some of the conclusions reported in the heatmap.

5/ Figure 6B: I find it confusing that the authors changed the order of the bar graph in the left and right panels. I would recommend the author to be consistent between the two.

6/ Figure S7B: report the data in "number of EVs / cells" instead of ul of EVs.

SIGNIFICANCE

Of note, the content of the section of the review "SIGNIFICANCE" was transmitted to the authors during the reviewing process. I will thus not repeat it again.

COMMENTS ON THE REVISED VERSION (OCT 21)

In their revised version, the authors took into consideration most reviewers' comments.

I will comment point by point on my matter of concern (although the authors were not aware of it at the time)

Point1: The authors did not address this point in their revised version. However, I believe that it would bring value for the study to quantify the copy number of mRNA per EV and comment on the comparison with miRNA copy number per EV. In addition, it is valuable for the study whether this could explain why mRNA, but not miRNA functional transfer, can be observed in their experimental setup.

Point2: Following reviewer 1's comment, the author successfully addressed my point of concern related to the sensitivity of their miRNA reporter assay. In the new figure S8, they could convincingly demonstrate a 50% reduction in luciferase activity with around 20-300 miRNA/cells, and therefore that they can detect the functional impact of low expressed miRNA.

Point 3: The authors did not address this point in their revised version. Please include these modifications in the following revised version.

Point 4. The authors did not address this point in their revised version. Please include these modifications in the following revised version.

Point 5: Following reviewer 3's comment, the authors modified Fig 6B to include additional baseline control for their experiment. It is an excellent improvement to their reported finding. Still, as initially commented, I would be happy that the authors keep the same order on the horizontal axis between mi.-BART1 and miR-BHRF-1-2 pannel.

Point 6: The authors did not address this point in their revised version. Please include these modifications in the following revised version.

**Have all data underlying the figures and results presented in the manuscript been provided?**

Reviewer #1: Yes

Reviewer #2: Yes

Reviewer #3: **No: **Although the author mentioned that they provided an xls file with raw numerical value that underlies their graph and statistics, I was not able to find them in the review package. Please check that they are indeed available.

PLOS authors have the option to publish the peer review history of their article (what does this mean?). If published, this will include your full peer review and any attached files.

Reviewer #1: **Yes: **Jun-An Chen

Reviewer #2: No

Reviewer #3: No

---

## [Editor Report · Decision Letter 2]

16 Nov 2021

Dear Dr Hammerschmidt,

We are pleased to inform you that your manuscript entitled "MicroRNAs are minor constituents of extracellular vesicles that are rarely delivered to target cells" has been editorially accepted for publication in PLOS Genetics. Congratulations!

We appreciated your effort to address the additional comments from reviewer 3, whose review was partially transferred to us from review commons. We suggest to take the proposed option B when addressing the first comment from reviewer 3. Please revise your manuscript to include this response to reviewer 3. 

Before your submission can be formally accepted and sent to production you will need to incoporcomplete our formatting changes, which you will receive in a follow up email. Please be aware that it may take several days for you to receive this email; during this time no action is required by you. Please note: the accept date on your published article will reflect the date of this provisional acceptance, but your manuscript will not be scheduled for publication until the required changes have been made.

Yours sincerely,

Lin He

Associate Editor

PLOS Genetics

John Greally

Section Editor: Epigenetics

PLOS Genetics

Comments from the reviewers (if applicable):

**Data Deposition**

http://datadryad.org/submit?journalID=pgenetics&manu=PGENETICS-D-21-00848R2

**Press Queries**

---

## [Editor Report · Acceptance letter]

1 Dec 2021

PGENETICS-D-21-00848R2 

MicroRNAs are minor constituents of extracellular vesicles that are rarely delivered to target cells 

Dear Dr Hammerschmidt, 

We are pleased to inform you that your manuscript entitled "MicroRNAs are minor constituents of extracellular vesicles that are rarely delivered to target cells" has been formally accepted for publication in PLOS Genetics! Your manuscript is now with our production department and you will be notified of the publication date in due course.

With kind regards,

Katalin Szabo

PLOS Genetics

On behalf of:
